# Implementation and effectiveness of non-specialist mediated interventions for children with Autism Spectrum Disorder: A systematic review and meta-analysis

Sadiq Naveed[1], Ahmed Waqas[2]*, Afshan Naz Amray[3], Raheel Imtiaz Memon[4], Nisma Javed[5], Muhammad Annas Tahir[6], Sherief Ghozy[7], Nusrat Jahan[8], Anum Saeed Khan[9], Atif Rahman[10]

1 Kansas University Medical Center, Kansas City, Kansas, United States of America, 2 Human Development Research Foundation, Rawalpindi, Pakistan, 3 Dow University of Health Sciences, Karachi, Pakistan, 4 Henry Ford Allegiance Health, Jackson, MI, United States of America, 5 Services Institute of medical sciences, Lahore, Pakistan, 6 Kingsbrook Jewish Medical Center, New York City, New York, United States of America, 7 Neurosurgery Department, El-Sheikh Zayed Specialized Hospital, Giza, Egypt, 8 Rush University Medical Center, Chicago, IL, United States of America, 9 Westchester Medical Center-New York Medical College, Valhalla, NY, United States of America, 10 University of Liverpool, Liverpool, England, United Kingdom

☯ These authors contributed equally to this work.
* ahmedwaqas1990@hotmail.com

**Data Availability Statement:** All relevant data are within the manuscript and its Supporting Information files.

## Abstract

### Introduction

In recent years, several non-specialist mediated interventions have been developed and tested to address problematic symptoms associated with autism. These can be implemented with a fraction of cost required for specialist delivered interventions. This review represents a robust evidence of clinical effectiveness of these interventions in improving the social, motor and communication deficits among children with autism.

### Methods

An electronic search was conducted in eight academic databases from their inception to 31st December 2018. A total of 31 randomized controlled trials were published post-2010 while only 2 were published prior to it. Outcomes pertaining to communication, social skills and caregiver-child relationship were meta-analyzed when reported in > 2 studies.

### Results

A significant improvement was noted in child distress (SMD = 0.55), communication (SMD = 0.23), expressive language (SMD = 0.47), joint engagement (SMD = 0.63), motor skills (SMD = 0.25), parental distress (SMD = 0.33) parental self-efficacy (SMD = 0.42) parent-child relationship (SMD = 0.67) repetitive behaviors (SMD = 0.33), self-regulation (SMD = 0.54), social skills (SMD = 0.53) symptom severity (SMD = 0.44) and visual reception (SMD = 0.29).

**Funding:** The authors received no specific funding for this work.

**Competing interests:** The authors have declared that no competing interests exist.

## Conclusion

Non-specialist mediated interventions for autism spectrum disorder demonstrate effectiveness across a range of outcomes for children with autism and their caregivers.

## Introduction

Autism Spectrum Disorder (ASD) is a neurodevelopmental disorder characterized by challenges in social communication and interaction and repetitive stereotypical behaviors, generally detectable in the first 2 years of life [1]. Although there are no definitive pharmacological treatments for the core deficits in ASD, several psychological interventions are used to address the communication and social skill deficits among children with the condition. Such interventions, if delivered in the early developmental period, can have long lasting positive impact on the lives of patients with ASD as well as their caregivers [1].

The Global Burden of Disease study was amongst the first to highlight the global prevalence of ASD, estimated at 52 million–a prevalence rate of 7.6 per 1000 –and disability adjusted life years estimated at 58 per 100,000 population [2,3]. The cost associated with interventions delivered by specialist mental health professionals is high, ranging from USD 40,000 to 80,000 per year, which is not feasible for low resource settings [4]. Therefore, the treatment gap associated with ASD is very high, especially in in low and middle countries (LMIC). For instance, a recent report from 14 African countries highlighted the lack of ASD services throughout Africa [5]. These statistics highlight the need for improved access to ASD services in LMIC as a priority from both public health as well as human rights perspectives.

To bridge these inequities in provision of treatment services for ASD, 'task sharing' strategies have been proposed to redistribute mental health services from specialists to non-specialist health workers [6]. In this context, there has been a recent focus on interventions that are delivered or mediated through non-specialists–parents, teachers, caregivers, and peers–that aim to improve developmental, educational, or behavioral outcomes among children with ASD. There are several advantages of non-specialist delivered interventions. For instance, parents, caregivers, and peers are part of the environment of children with ASD, thus providing ample opportunities for incidental therapeutic contacts that, if effective, can lead to a positive impact. The involvement of these stakeholders, including parents, teachers, caregivers, and peers, can be key to making these interventions acceptable and accessible for individuals with ASD [7].

A recent systematic review of single case studies highlighted several non-specialist mediated treatment programs such as the *SENSE Theater*, *LEAP intervention*, *Pivotal response treatment (PRT)*, *Social Stories TM*, *and video modelling* [8]. These interventions focus mainly on improving behavioral patterns, deficits in vocabulary and expressive language and social communication among children with ASD. The costs associated with these treatments is significantly lower than that of specialist delivered interventions, for instance, one effective program, the play project home consultation program, cost 3500 to 4500 USD per year per child compared to 40,000 to 80,000 USD per year for treatments delivered by specialists [9].

While the aforementioned treatment programs differ in their content, all aimed to improve one or more core deficits of ASD. For instance, Social Stories TM are stories written in first person from the perspective of the target individual engaged in a particular social situation, and explaining the behavior expected in it [8]. Peer mediation trains children with autism and their peers to interact during social engagement, and hence, improve social skills, joint

attention and engagement and communication [10]. SENSE Theater involves children with autism and their peers in theaters aiding in an understanding of socially expected behaviors. Video modelling involves videos depicting individuals demonstrating expected behavior to the children with autism [11]. And pivotal response treatment (PRT) trains children in pivotal behaviors required for daily functioning, such as social initiations and responsivity, self-efficacy, and motivation [12].

There is a paucity of comprehensive systematic reviews and meta-analysis of randomized controlled trials detailing content of different interventions, their effectiveness across different outcomes and quality of available evidence. Therefore, the present review was designed to, a) assess the effectiveness of non-specialist delivered or mediated interventions in ASD; b) systematically evaluate relevant implementation processes involved in these non-specialists delivered interventions for ASD, and c) and to rate the quality of evidence across different outcomes using the World Health Organization's recommended Grading of Recommendations Assessment, Development and Evaluation (GRADE) criteria (described below).

## Methods

This review was conducted as per the updated PRISMA guidelines [13] (S1 File), and the protocol registered in PROSPERO (CRD42017066009).

### Search process & selection criteria

An academic search was conducted in eight electronic databases including PubMed, Scopus, Web of Science, POPLINE, New York Academy of Medicine, PsycINFO, Psycharticles, and CINAHL, from their inception to 31st December 2018, using following search terms (S2 File):

(("autism spectrum disorder" OR Asperger OR autis* OR "pervasive developmental disorder" OR "childhood disintegrative disorder") AND (intervention OR treatment OR RCT OR trial) AND (parent-mediated OR parent-delivered OR "non-specialist mediated" OR "non-specialist delivered" OR teacher-mediated OR "teacher delivered" OR "aide delivered" OR "aide mediated" OR "peer delivered" OR "peer mediated")). No restrictions or database filters regarding language, time period or publication year were applied.

Three independent reviewers screened the aforementioned databases for eligible studies based on their titles and abstracts, followed by screening of full texts. All discrepancies among reviewers were resolved through discussion between reviewers and senior authors. All studies were assessed for eligibility against following criteria:

### Inclusion criteria

1. Individuals with a clinical diagnosis of ASD, screened for ASD using questionnaires or clinician diagnosis, Asperger's syndrome, and childhood disintegrative disorder were included.

2. Only studies assessing the efficacy through randomized controlled trials were included.

3. No restriction to age, gender, language, country, socioeconomic status or time period was applied.

4. Studies focusing on the parent, caregiver, peer, teacher or any other non-specialist mediated or delivered interventions were included.

## Exclusion criteria

1. Overlapping data sets reporting results from same study.

2. Studies which are not randomized controlled trials will be excluded.

3. Books, conference papers, theses, editorials, case reports, case series, reviews and articles without available full text will be excluded.

4. Specialist-delivered Interventions.

5. Non-original articles (reviews and analyses)

6. In Vitro studies and non-human trials.

7. Interventions conducted among adults with ASD were excluded.

## Data extraction, risk of bias assessment & GRADE evidence

All data were extracted independently by three teams of reviewers using manualized data extraction forms and any disagreements among the reviewers, were resolved through discussion in conjunction with a senior author.

Data pertaining to participant characteristics, study setting, nature of intervention and outcomes will be extracted. For outcomes, an apriori decision was taken to include all types of psychometric testing whether conducted by specialists, teachers or parents. A variety of psychometric scales used for measurement of symptoms of autism are reported in the literature. We conducted a thorough audit of included studies to identify the psychometric scales used and categorized them under a unifying category. For instance, total symptom severity comprised of several scales such as Autism Diagnostic Observation Schedule; Autism Behaviour Checklist; Vineland Adaptive Behaviour Scale and Childhood Autism Rating Scale among others.

If there was a trial with more than one publication, preference was given to the primary publication. A US board certified child psychiatrist also devised a taxonomy form for active ingredients of interventions with detailed instructions regarding content, strategies and elements of interventions. Moreover, two authors assessed the quality of the studies without blinding to authorship or journal, using The Cochrane tool for randomized controlled trials, against several matrices: a) sequence generation, b) allocation concealment, c) blinding of participants and personnel, d) blinding of outcome assessment, e) incomplete outcome data, f) selective reporting and g) other bias" [14].

The meta-analytical evidence in present review was assessed for its quality using the recommendations outlined by the Grading of Recommendations, Assessment, Development and Evaluations (GRADE) Working Group [15]. These recommendations allow for rating of meta-analytical ranging from high to very low based on its study design, risk of bias, inconsistency, indirectness in targeted population and suitability of intervention, imprecision, publication bias and magnitude of effect size [15]. The evidence is judged across outcomes, where each concern in aforementioned matrices is rated as serious or very serious, stepping down the quality of evidence by one or two levels respectively [15].

## Data analysis

Descriptive statistics pertaining to characteristics of the study and implementation processes including elements of interventions were explored using Microsoft Excel. Thereafter, using Comprehensive meta-analysis software, a series of meta-analyses were run for similar

outcomes assessed post-intervention or primary time points [16]. Only those outcomes were introduced in the meta-analysis that were reported in ≥ 2 studies. Studies reporting similar outcomes were pooled together, weighted by employing inverse variance method, thus, estimating pooled effect sizes expressed as standardized mean differences with 95% confidence intervals (CI) [16]. Depending on the extent of heterogeneity, data were pooled together using either the fixed or random effects. Heterogeneity was considered significant at a cut off value ≥ 40%. However, we applied random effects analysis for all of the outcomes because of heterogeneity in assessment of outcomes across included studies [17]. Sensitivity analyses was conducted by excluding individual studies individually to ascertain their effects on the pooled effect size. When ≥ 5 studies reported an outcome, publication bias was assessed for asymmetry by visualizing the Begg's funnel plot and Egger's regression statistics (P ≤ 0.10) [18]. Pooled effect sizes were then adjusted for publication bias using Duvall & Tweedie's Trim and Fill method [19]. To ensure an appropriate statistical power, when there were ≥4 studies reporting an outcome among different groups, subgroup analyses were conducted. Lastly, an outcome reported in ≥ 10 studies allowed meta-regression analyses to identify potential moderators of intervention effects among children with autism [20].

## Results

Searching of academic databases yielded a total of 659 non-duplicate references to be screened based on their titles and abstracts. Out of these, 596 citations were excluded, retaining 63 full texts. Thereafter, 33 randomized controlled trials were deemed eligible after screening of these full texts against inclusion and exclusion criteria. Detailed results have been presented in PRISMA flowchart (Fig 1). S3 File presents effect sizes, means (SD) and subgroups for individual studies.

### Study characteristics

A total of 31 studies were published post-2010 while only 2 were published prior to it [21,22]. Only 2 studies was conducted in a low and middle income country [6] (Divan, 2018) while others were conducted in high income countries including USA (n = 21), Australia (n = 3), UK (n = 2), Canada (n = 2) and 1 each in Belgium, Norway, and Netherlands. Only two of the studies was a cluster randomized control trial while rest were individual RCTs [23] (Morgan et al, 2018). National Institutes of Health were the major funder of these trials (n = 11). A total of 24 studies were conducted in urban areas and 3 in both rural and urban areas (missing n = 3). Table 1 provides further details on these variables.

### Intervention characteristics

Most of the interventions took place in the community (n = 9) and educational settings, (n = 10) followed by home (n = 7), healthcare setting (n = 2), videoconferencing (n = 1), and the rest in mixed settings (n = 3). The majority of the interventions (n = 14) had employed the Autism Diagnostic Observation Schedule (ADOS) for screening of children with autism. Other scales such as Childhood Autism Rating Scale (CARS), Modified Checklist for Autism in Toddlers (M-CHAT) and Autism Behavior Checklist (ABC), Checklist for Autism Spectrum Disorder (CASD) were also employed (Fig 2 and Fig 3).

The children were assessed for inclusion by a variety of professionals including research personnel (n = 13), psychologists/therapists (n = 5), multidisciplinary child and adolescent mental health (CAMH) team (n = 4), teachers (n = 1), certified intervention providers (n = 2); the information was unavailable for 5 studies. Delivery agents of interventions included parents (n = 17), peers (n = 4), and school staff (n = 3). These delivery agents were trained by

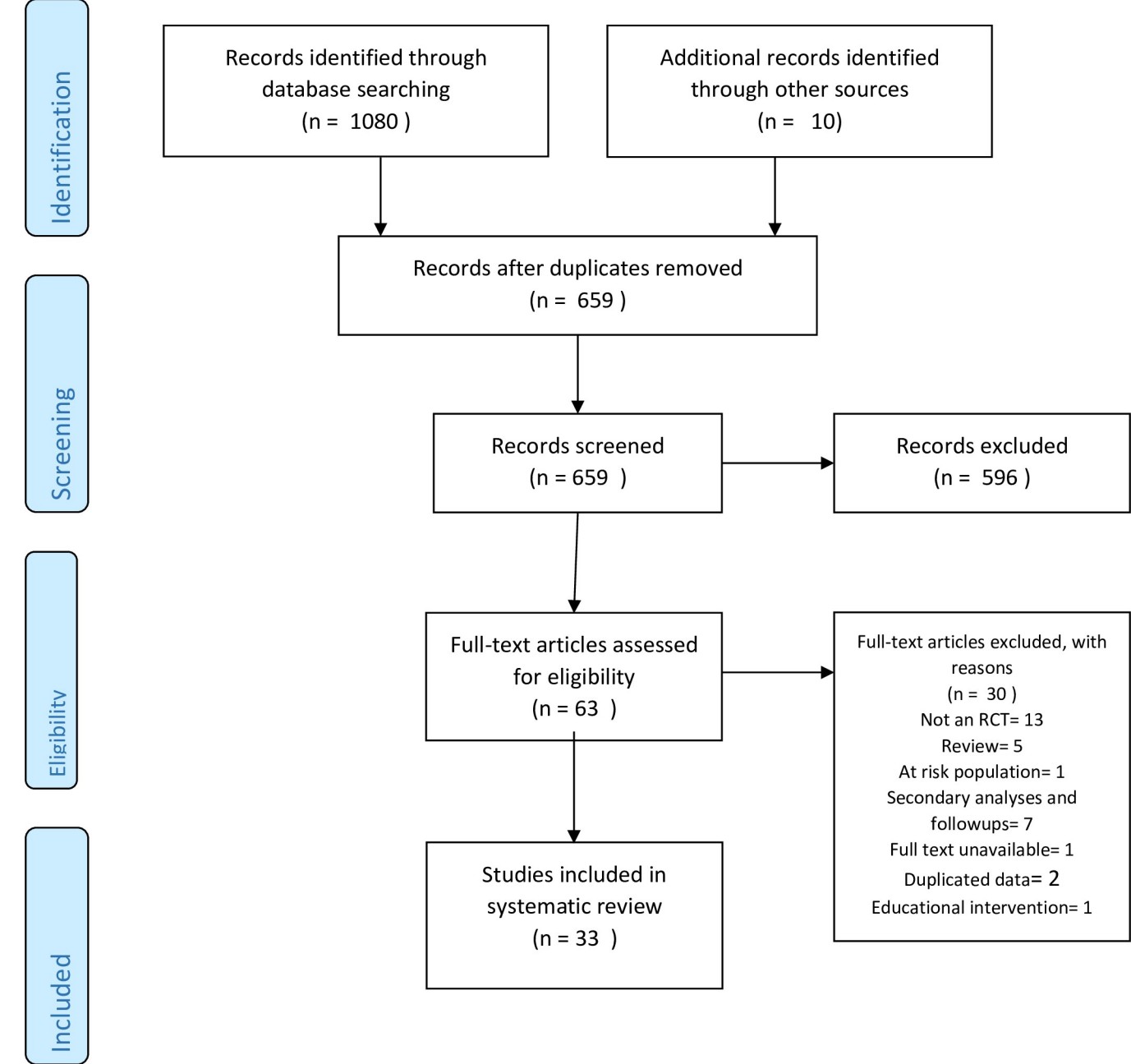

**Fig 1. PRISMA flow diagram demonstrating study selection process.**

trained certified interventionists (n = 6), researchers (n = 6), therapists (n = 5), speech pathologists (n = 1), trained graduates and doctoral students (n = 6), massage trainer (n = 1), local specialist health workers (n = 2), multidisciplinary counselors (n = 3), music therapists (n = 1), and this information was missing for two studies. Competency evaluation was performed in 3 studies [24,25]. Fidelity was not rated in 7 of the studies [21,22,26–29]. None of the trials reported provision of any certification or remuneration to the delivery agents except one study [30]. Supervision of delivery agents was done onsite (n = 23), onsite and online (n = 2), onsite and videotaped (n = 3), while this information was not available for one study [31]. A majority

**Table 1. Intervention characteristics.**

| Study | Country of study | age range of respondents | Study Design | Geographical scope | Setting of intervention |
|---|---|---|---|---|---|
| Cook et al, 2017 [27] | AUSTRALIA | 4 years to 6 years | RCT | Urban | Griffith University Psychology Clinic |
| Corbett et al, 2016 [26] | USA | 8 years to 14 years | RCT | Urban | School |
| Corbett et al, 2017 [36] | USA | 8 years to 14 years | RCT | Urban, | Theater/ Home |
| Green et al, 2010 [35] | UK | 2 years to 4 years and 11 months | RCT | Urban | premises of local primary care trusts |
| Ingersoll et al, 2016 [31] | USA | 19 months and 3 months | RCT | Urban | Video conference/online |
| Kasari et al, 2015 [46] | USA | 22 months—36 months | RCT | Urban | Community |
| Kasari et al, 2012 [34] | USA | 6 years—11 years | RCt | Urban | School |
| Rahman et al, 2016 [6] | India & Pakistan | 2–9 years | RCT | Urban | One-to-one clinic or home sessions between the health worker and the parent with the child present. All sessions in India were delivered in the home, and all those in Pakistan in the clinic |
| Roeyers, 1996 [22] | Belgium | 5 and 13 years old. | RCT | NA | Playing sessions took place in a playroom at the school or institution |
| Schertz, 2013 [30] | U.S.A. | under 30 months | RCT | Rural and urban | Homes |
| Shire, 2016 [39] | U.S.A. | 36 months | RCT | Urban | Community |
| Siller et al, 2014 [37] | U.S.A. | 2–6 years | RCT | Urban | Community |
| Silva et al, 2011 [25] | U.S.A. | 3–6 years | RCT | Urban | Community |
| Silva et al, 2015 [24] | U.S.A. | 2–5 years | RCT | urban | Community |
| Solomon et al, 2014 [9] | USA | 2 yr 8 mo–5 yr 11 mo | RCT | Not Mentioned | |
| Strain et al, 2011 [23] | USA | preschoolers with asd | Cluster RCT | Urban, SEMI URBAN AND RURAL ALL THREE | schools |
| Venker et al, 2012 [32] | USA | 41 MONTHS | RCT | Not Mentioned | Community |
| Kamps et al, 2015 [41] | USA | kindergarten age group (3 yrs) | RCT | URBAN | School |
| Thompson et al, 2014 [28] | Australia | 3 to 6 years | RCT | Urban | Home |
| Grahame et al, 2015 [42] | ENGLAND, UK | 3 to 7 years | RCT | Urban | Community |
| Carter et al, 2011 [33] | USA | 20.25 months | RCT | Urban | Community clinics |
| Kaale et al, 2012 [51] | NORWAY | 24 to 60 months | RCT | Urban | School |
| Jocelyn et al, 1998 [21] | CANADA | 24 to, 72 months | RCT | Urban | Community |
| Poslawsky et al, 2015 [29] | NETHERLANDS | childrens' age:16 to 61 months; parents age : 25 to 52 years | RCT | urban | hospital and home |
| Brian et al, 2017 [47] | Canada | 16–30 months | RCT | Urban | Home |
| Divan et al, 2019 [48] | India | 27–105 months | RCT | Rural | Home |
| Ibanez et al, 2018 [49] | USA | N/A | RCT | Urban | Home |
| Kuravackel et al, 2017 [50] | USA | 3 to 12 years old | RCT | Rural and urban | University, regional health center, clinic |

*(Continued)*

**Table 1.** (Continued)

| Study | Country of study | age range of respondents | Study Design | Geographical scope | Setting of intervention |
|---|---|---|---|---|---|
| Matthews et al, 2018 [51] | USA | 13 to 17 years old | RCT | Urban | Community-based non-profit autism center |
| Morgan et al, 2018 [52] | USA | Mean age = 6.79 years | cRCT | Urban | School |
| Parsons et al, 2018 [53] | Australia | 2 to 6 years old | RCT | Rural | Home |
| Vernon et al, 2018; Ko et al, 2018 [54,55] | USA | 12 to 17 years | RCT | Urban | School |

of the trials (n = 21) were standalone interventions while rest of them were integrated with school curriculum (n = 1), existing services (n = 3) or speech, language and occupational therapy (n = 1) [22,28,32]. Psychopharmacological treatment was included in one trial [6]. These variables are reported in greater detail in Table 2.

**Fig 2. Summary effect sizes for symptom severity.**

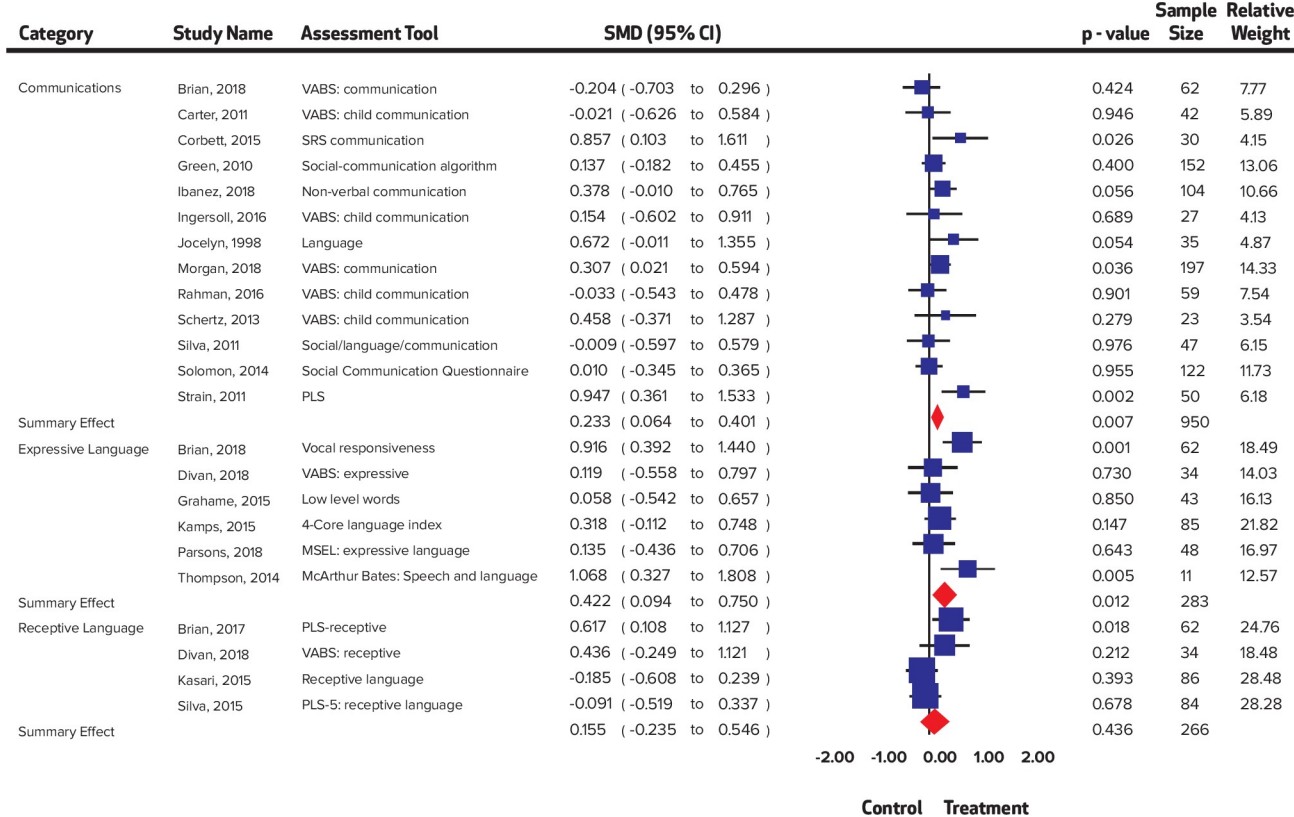

**Fig 3. Summary effect sizes for communication skills.**

## Density of dosage

The mean number of sessions was 53.29 (SD = 158.07), ranging from 5 [29] to 834 [23] sessions, while this information was not reported by two studies [33]. Duration of each session was averaged at 130.63 minutes (SD = 281.47) and ranged from 10 minutes (daily) to 1500 minutes per week [26,27], and was not reported by two studies[33]. Mean duration for program was 22.62 weeks (SD = 33.34), lasting a minimum of 6 weeks [34] and maximum of 182.50 weeks [23]. Booster sessions were conducted in a total of six trials [26,27,29,35] (Brian and Matthews). These variables are reported in greater detail in Table 2.

## Strategies & elements of interventions

The content of the included interventions differed in their theoretical orientation: Cooka et al (2017) employed cognitive behavioral strategies (CBT) [27]; Social Emotional NeuroScience Endocrinology (SENSE) theater (sensory stimulation and creative techniques including theater) [26,36]; family mediated Preschool Autism Communication Trial (PACT) and Parent-mediated intervention for Autism Spectrum Disorders in South Asia (PASS) program (pragmatic language development) [6,35]; Project Impact (Communication/Pragmatic language) [31]; Peer interventions (Communication/pragmatic language intervention)[31,34]; Qigong Sensory Treatment (QST) home program (Communication/pragmatic language development) [37]; Qigong massage (sensory stimulation) [24,25]; Joint Attention, Symbolic Play, Engagement, and Regulation programme (JASPER) (behavioral/pragmatic language & communication)[38–40]; Play project (behavioural, pragmatic language and communication)[9]; LEAP

**Table 2. Strategies employed in interventions.**

| Author, year | No. of sessions | Duration (minutes) | Program duration (weeks) | Delivery agent | Training provider | Name of intervention | Theoretical orientation of intervention | Strategies employed |
|---|---|---|---|---|---|---|---|---|
| Cook et al, 2017 [27] | 9 | 90 | 13 | Parent | Therapists | CBT | Behavioral | Psychoeducation Preventive Strategies related to environmental and parenting. Development of an exposure hierarchy and graded exposure. Affective education. Emotion regulation skills training. |
| Corbett et al, 2016 [26] | 10 | 240 | 10 | Peer | Researchers | Social Emotional NeuroScience Endocrinology (SENSE) Theater | Sensory stimulation and creative techniques including theater | Observing, interpreting, and articulating thoughts and feelings; Theatrical techniques such as improvisation, role-playing, scripted interaction, video modeling, and performing; behavioral techniques to address |
| Corbett et al, 2017 [36] | 10 | 240 | 8 | Peer | NR | Social Emotional NeuroScience Endocrinology (SENSE) Theater | Same as above | |
| Green et al, 2010 [35] | 12 | 120 | 24 | Parent | Speech and language pathologist | Preschool Autism Communication Trial (PACT) | Pragmatic language | Establishing shared attention Synchronicity and sensitivity Focusing on language input Establishing routines and anticipation Increasing communication |
| Ingersoll et al, 2016 [31] | 48 | 105 | 24 | Parents/ Therapists assisted | Masters' level therapists | ImPACT Online website | Communication/ Pragmatic language | Social engagement Language development Social imitation Play |
| Kasari et al, 2015 [46] | 20 | 60 | 10 | Caregivers/ parents. | Trained interventionist | JASPER | Behavioral, pragmatic language | Joint attention Symbolic play Social engagement Emotional and behavioral Regulation |
| Kasari et al, 2012 [34] | 12 | 10 | 6 | Peer | Education psychologists graduates | Peer-mediated (PEER) intervention | Pragmatic language | Social interaction Positive peer modelling |
| Rahman et al, 2016 [6] | 12 | 60 | 24 | Parent | Local specialist, health worker | PASS (Same as PACT) | Pragmatic language | Establishing shared attention Synchronicity and sensitivity Focusing on language input Establishing routines and anticipation Increasing communication |
| Roeyers, 1996 [22] | 15 | 23.33 | NR | Peer | NR | Peer mediated intervention | Pragmatic language | Focusing on faces Social reciprocity Initiating Joint attention Responding to joint attention |
| Schertz, 2013 [30] | 16 | 40 | 16 | Parents | Intervention coordinators | Joint Attention Mediated Learning | Communication/ Pragmatic language | Joint attention/engagement |
| Shire, 2016 [39] | 10 | 60 | 10 | Parent | Trained clinician | JASPER | Communication/ Pragmatic language | Joint attention/engagement |

(*Continued*)

**Table 2.** (Continued)

| Author, year | No. of sessions | Duration (minutes) | Program duration (weeks) | Delivery agent | Training provider | Name of intervention | Theoretical orientation of intervention | Strategies employed |
|---|---|---|---|---|---|---|---|---|
| Siller et al, 2014 [37] | 12 | 90 | 12 | Parent | Trained graduates and post-doctoral students | QST Home Program | Communication/ Pragmatic language | Psychoeducation Responsive parental communication/ Maternal synchronization Response to joint attention Communication Play Eliciting imitation Eliciting eye contact. |
| Silva et al, 2011 [25] | 7 | 15 | 7 | Parent | Massage trainer | Qigong massage treatment | Sensory stimulation | Qigong massage |
| Silva et al, 2015 [24] | 20 | 15 | 20 | Parent | Therapists | Qigong massage treatment | Sensory stimulation | Qigong massage |
| Solomon et al, 2014 [9] | 12 | 180 | 48 | Parent | Multidisciplinary team | PLAY Project Home Consultation Intervention Program | Communication, behavioral, Pragmatic language | Self regulation Social Engagement Communication Shared Meanings and Symbolic Play Emotional Thinking |
| Strain et al, 2011 [23] | 834 | 180 | 182.5 | Teacher | Trained staff | LEAP (Learning Experiences and Alternative Program for Preschoolers and Their Parents) | Creative, Pragmatic language, behavioral | Organization and planning Communication skills Social interactions Positive behavioral guidance Interactions with children and families |
| Venker et al, 2012 [32] | 21 | 75 | 10 | Parents | Graduate Students | More Than Words: The Hanen Program for Parents of Children with Autism Spectrum Disorder | Communication/ Pragmatic language | Non-verbal and verbal communication, prompts |
| Kamps et al, 2015 [41] | 97 | 27.5 | 24 | school staff members | researchers trained school staff members | Peer Networks Intervention Procedures | Communication | Social interaction Communication |
| Thompson et al, 2014 [28] | 16 | 35 | 16 | Parents | Music therapists | Family-centred music therapy (FCMT) | Creative (music), Pragmatic language | Social engagement, shared attention, joint attention |
| Grahame et al, 2015 [42] | 8 | 120 | 8 | Parents | Early Year Professionals | The Managing Repetitive Behaviours Programme (MRBÓ) | Behavioral | Psychoeducation, behavioral |
| Carter et al, 2011 [33] | NR | NR | NR | Parents | Researchers | Hanen's 'More Than Words' | Pragmatic language | Social Interaction |
| Kaale et al, 2012 [51] | 80 | 20 | 8 | Teachers | Counselors with a degree in special education, psychology or social sciences | Modified JASPER intervention: Preverbal pragmatic language Introduction and responsiveness Nonverbal comm. | Pragmatic language | Joint attention/social engagement |
| Jocelyn et al, 1998 [21] | 10 | 180 | 12 | Parent and child care worker | Autism behavioral specialist, child development counselor, community family services workers | Autism Preschool Program | Behavioral | Behavioral |

*(Continued)*

**Table 2.** (Continued)

| Author, year | No. of sessions | Duration (minutes) | Program duration (weeks) | Delivery agent | Training provider | Name of intervention | Theoretical orientation of intervention | Strategies employed |
|---|---|---|---|---|---|---|---|---|
| Poslawsky et al, 2015 [29] | 5 | 75 | 12 | Parents | Researchers | Video-feedback Intervention to promote Positive Parenting adapted to Autism (VIPP-AUTI) | Pragmatic language, behavioral | Mastery motivation and child playParent-child interaction Joint attentionRecognition of children's affect and emotions |
| Brian et al., 2017 [47] | 13 | 90 | 12 | Parents | Researchers, clinicians | Social ABCs | Communication, behavioral | The ABCs of learning, enhancing Communication, sharing positive emotion, motivation and arousal, play and the Social ABCs, daily care-giving activities, managing behavioral challenges, and taking care of yourself |
| Divan et., 2018 [48] | 12 | 17.5 | 24 | Parents | Researchers | Parent mediated intervention for Autism Spectrum Disorder Plus" (PASS Plus) | Pragmatic language, behavioral, sensory stimulation | Increased parental synchronous responses, increased understanding of child's verbal and non-verbal responses as part of PASS. Plus module involved psycho-education and assessment of the most disruptive comorbidity for the family. It included strategies for sensory seeking and sensory defensive behaviors. The behavioral challenges focused on identifying reasons for hyperactivity, self-harming, and aggression. Additional strategies targeted sleep problems, bed wetting issues, toileting difficulties, restricted diet, pica, and inflexible routines. Parental well-being was also addressed. |
| Ibanez et al., 2018 [49] | - | - | 18 | Parents | Not mentioned | Enhancing Interactions Tutorial | Communication, behavioral | This tutorial educated parents about definition of home routine, their importance, and tips for starting and ending home routines. It also enhanced awareness about challenges for children with ASD and increase their engagement in routine by assessing current level of participation. The parents were taught about using choice boards, first-ten boards, visual schedules, timers, prompting, reinforcement, imitation, and language understanding. |

(*Continued*)

**Table 2.** (Continued)

| Author, year | No. of sessions | Duration (minutes) | Program duration (weeks) | Delivery agent | Training provider | Name of intervention | Theoretical orientation of intervention | Strategies employed |
|---|---|---|---|---|---|---|---|---|
| Ko et al., 2018 [55] | 20 | 90 | 20 | Peers | Therapists | Social Tools And Rules for Teens socialization (START) intervention | Communication | Unstructured socialization with peers and facilitators, social immersion, self-management of skills, role play, active discussion and practice, structured games, and developing social goals for next week. |
| Kuravackel et al., 2018 [50] | 8 | 90 | 8 | Parents | Therapists | COMPASS for Hope (C-Hope) | Pragmatic language, communication | Psychoeducation to parents, assessment of problematic issues, education on principles of behaviors and learning, teaching positive behavioral approaches, importance of environmental support, preparation and review of individual behavioral plan for children. |
| Matthews et al., 2018 [51] | 14 | 90 | 14 | Peers | Certified PEER providers | PEERS curriculum | Communication | Initiation of peer interactions, behavioral rehearsals, and modeling of appropriate social skills by facilitators. |
| Morgan et al., 2018 [52] | 32 | 1500 (weekly) | 32 | Teachers | Certified coaches | Communication, Emotional Regulation, and Transactional Support (SCERTS) Intervention | Communication, behavioral | Assessment of individual's language stage and selection of goals and objectives. The targeted activities were planned to address these goals through direct teaching as needed, guided practice with feedback, teacher practice and reflection with feedback, and teacher independence. |
| Parsons et al., 2018 [53] | 90 | 20 | 12 | Parents | Researchers | Therapeutic Outcome By You (TOBY) application | Communication, pragmatic language, and sensory | Selection of activities based on a curriculum tree and uses principles of Applied Behavioral Analysis (ABA) for skill attainment by identifying problems and techniques to change environment. |
| Vernon et al., 2018 [54] | 20 | 90 | 20 | Parents | Therapists | Social Tools And Rules for Teens socialization (START) intervention | Communication | Unstructured socialization with peers and facilitators, social immersion, self-management of skills, role play, active discussion and practice, structured games, and developing social goals for next week. |

project i.e. Learning Experiences and Alternative Program for Preschoolers and Their Parents (behavioral, creative, pragmatic language training) [23]; Hanen's "more than words" intervention program (behavioral, pragmatic language) [32,33]; Peer network intervention procedure (communication) [41]; family centered music therapy (Creative and pragmatic language training)[28]; The Managing Repetitive Behaviours Programme (Behavioural and psychoeducational) [42]; psychoeducation program autism preschool program (behavioural & psychoeducation) [21] and the Video-feedback Intervention to promote Positive Parenting

(behavioural & pragmatic language intervention) adapted for Autism by Poslawsky et al [29]; Social ABCs (communication, behavioral); PASS plus (pragmatic language, behavioral and sensory stimulation); enhancing interactions tutorial (communication, behavioral); Social Tools And Rules for Teens socialization (START) intervention (communication); COMPASS for Hope (communication); PEERS curriculum (communication); Therapeutic Out-come By You (TOBY) application (communication, pragmatic language and sensory stimulation). These variables are reported in Table 2.

Cook et al., (2017) was the sole RCT reporting the effectiveness of CBT based intervention programs among the children with autism [27]. It focused on psychoeducation, assessment, recognition and understanding of affect and cognitive schema, CBT based coping and relaxation exercises. Corbet et al., (2015, 2016) used SENSE theater technique targeting social skills [26,36]. Green et al (2010) and Rahman et al (2016) employed speech and language therapists in parent-mediated intervention to in elicit an improvement in communication skills among children [6,35]. Ingersoll et al, (2016), in her Project ImPACT used interactive and direct techniques to increase the ability of the child to engage and socially and improve their language skills respectively [31]. Kasari et al (2012), Roeyers (1996) and Kamps et al (2015) tested peer delivered intervention to improve social support, engagement, social interaction, play and conflict resolution skills among children with autism. JASPER model was tested for effectiveness in three studies [38–40]. Thompson et al., focused on family centered music therapy to improve initiation and responsive joint attention among children with autism [20]. Jocelyn (1998) et al delivered psychoeducation [21] and Poslawsky et al (2015) employed video recording of play situations and a mealtime to promote Positive Parenting adapted to autism [29]. Venker et al (2012) and Carter et al (2011) in their Hanen's "more than words" intervention employed child-oriented interaction promoting and language modelling strategies among children with autism [32, 33]. Grahame et al (2015) improved repetitive behaviors using techniques such as psychoeducation, reinforcement, planning and distraction [42]. Silva et al (2011 and 2015) tested the efficacy of Qigong massage treatment [24, 25], Siller et al (2014) employed QST home program to improve responsive parental behaviors [37]. Several other programs such as Joint Attention Mediated Learning (JAML) by Scher, 2013, encouraging opportunities for social interactions [30]. Strain & Edward (2011) tested a LEAP program (Learning Experiences and Alternative Program for Preschoolers and Their Parents) using a naturalistic approach to learning of social interaction [23]. Solomon et al's (2014) trained children in shared attention, self-regulation, engagement, initiating simple and complex communication using Coaching, modeling, video-feedback [9].

## Outcomes

The included trials revealed a number of outcomes including adaptive behaviors (6 trials, n = 286), child anxiety (2 trials, n = 42), child distress (2 trials, n = 76), communication and language (15 trials, n = 896), joint attention (7 trials, n = 464), joint engagement (4 trials, n = 261), motor skills (5 trials, n = 304), parental distress (7 trials, n = 441), parental self-efficacy (4 trials, n = 166), parent child relationship (6 trials, n = 372), repetitive behaviors (2 trials, n = 195), self-regulation (3 trials, n = 175), social skills (10 trials, n = 545), symptom severity (7 trials, n = 398), visual reception (3 trials, n = 198). A variety of psychometric instruments were utilized in the included studies, posing methodological heterogeneity in measurement of outcomes. The most commonly employed psychometric scales included Vineland Adaptive Behavior Scale, Mullen Scales for Early Learning, Autism Diagnostic Observation Schedule, Social Communication Questionnaire and Autism Behavior Checklist. For the purpose of meta-analysis, we combined effect sizes on all types of outcomes reported by teachers, parents or experts.

A significant improvement was noted in child distress (SMD = 0.55, 95% CI = 0.25 to 0.85, $I^2$ = 0%; $Chi^2$ = 1.76); communication (SMD = 0.23, 95% CI = 0.03 to 0.42 $I^2$ = 37.96%; $Chi^2$ = 17.73); expressive language (SMD = 0.47, 95% CI = 0.22 to 0.72 $I^2$ = 53.59%; $Chi^2$ = 8.62); joint engagement (SMD = 0.63, 95% CI = 0.21 to 1.06 $I^2$ = 75.88%; $Chi^2$ = 24.87); motor skills (SMD = 0.25 95% CI = 0.02 to 0.48 $I^2$ = 0%; $Chi^2$ = 4.18); parental distress (SMD = 0.33, 95% CI = 0.09 to 0.57 $I^2$ = 52.01%; $Chi^2$ = 18.75); parental self-efficacy (SMD = 0.42, 95% CI = 0.23 to 0.62 $I^2$ = 0%; $Chi^2$ = 4.64); parent-child relationship (SMD = 0.67, 95% CI 0.23 to 1.10 $I^2$ = 76.0%; $Chi^2$ = 20.83); repetitive behaviors (SMD = 0.33, 95% CI = 0.05 to 0.62 $I^2$ = 0%; $Chi^2$ = 0.17); self-regulation (SMD = 0.54, 95% CI = 0.06 to 1.03 $I^2$ = 55.91%; $Chi^2$ = 4.36); social skills (SMD = 0.53, 95% CI = 0.34 to 0.73 $I^2$ = 48.59%; $Chi^2$ = 31.12); symptom severity (SMD = 0.44, 95% CI = 0.27 to 0.60 $I^2$ = 0%; $Chi^2$ = 5.42) and visual reception (SMD = 0.29, 95% CI = 0.01 to 0.57 $I^2$ = 0%; $Chi^2$ = 1.22), while no significant improvement was noted in adaptive behaviors (SMD = 0.26, 95% CI = -0.001 to 0.52, $I^2$ = 41.44%; $Chi^2$ = 10.25); receptive language (SMD = 0.16, 95% CI = -0.24 to 0.55 $I^2$ = 53.34%; $Chi^2$ = 7.38); and joint attention (SMD = 0.16, CI = -0.22 to 0.54, $I^2$ = 76.13%; $Chi^2$ = 29.32). Forest plots are presented (Fig 2, Fig 3, Fig 4, Fig 5, Fig 6) and complete dataset has been provided as S3 File.

Sensitivity analyses revealed that removal of specific trials led to significant effect sizes pertaining to adaptive living (Silva, 2015 and Rahman, 2016), motor skills (Grahame et al; Solomon et al) and visual reception (Parsons et al).

## Moderator analyses

Initially, meta-regression analysis was run inclusive for all outcomes. It did not reveal any significant effects of age, year of publication or duration of program and session or number of sessions or quality of trials on the significance of these interventions. Meta-regression plots have been presented as S4 File. Subgroup analyses was run when specific outcomes reported in ≥ four studies. It did not reveal any significant differences among interventions delivered by different agents on outcomes of symptom severity and joint attention. While significant subgroup differences were observed in reporting of joint engagement with parent mediated interventions reporting highest effect sizes (Table 3).

## Quality rating & strength of evidence

Significant publication bias was revealed in reporting of social skills and symptom severity outcomes (Eggers statistics, P < 0.1). However, adjusted effect sizes for Social skills SMD = 0.42 (0.30 to 0.55) and symptom severity 0.38 (0.22 to 0.54) remained statistically significant (S4 File).

Cochrane's tool for risk of bias assessment among the included trials revealed an overall low risk of bias among majority of the studies. Random sequence generation was at a high/ unclear risk of bias among 8 trials, allocation concealment (n = 13). Frequency of studies reporting a high risk across other domains of Cochrane risk of bias tool were: Blinding of outcome assessors (n = 14), other sources of bias (n = 9), attrition bias (n = 8), selective reporting (n = 4) and blinding of participants and personnel (n = 0). A total of 11 studies were rated as having as having a high risk of overall bias i.e. ≥ 3 matrices of risk of bias tool were rated as having unclear or high risk of bias for these studies [22,23,26,27,32,41] (Fig 7 and S4 File). Fig 7 presents a clustered bar chart exhibiting frequencies of high, unclear and low risk bias across all matrices of Cochrane risk of bias tool. S4 File presents study wise risk of bias across all matrices of Cochrane risk of bias tool.

According to the GRADE criteria, evidence for four outcomes was rated as: High for communication skills, expressive language, motor skills, repetitive behaviors, and parental distress.

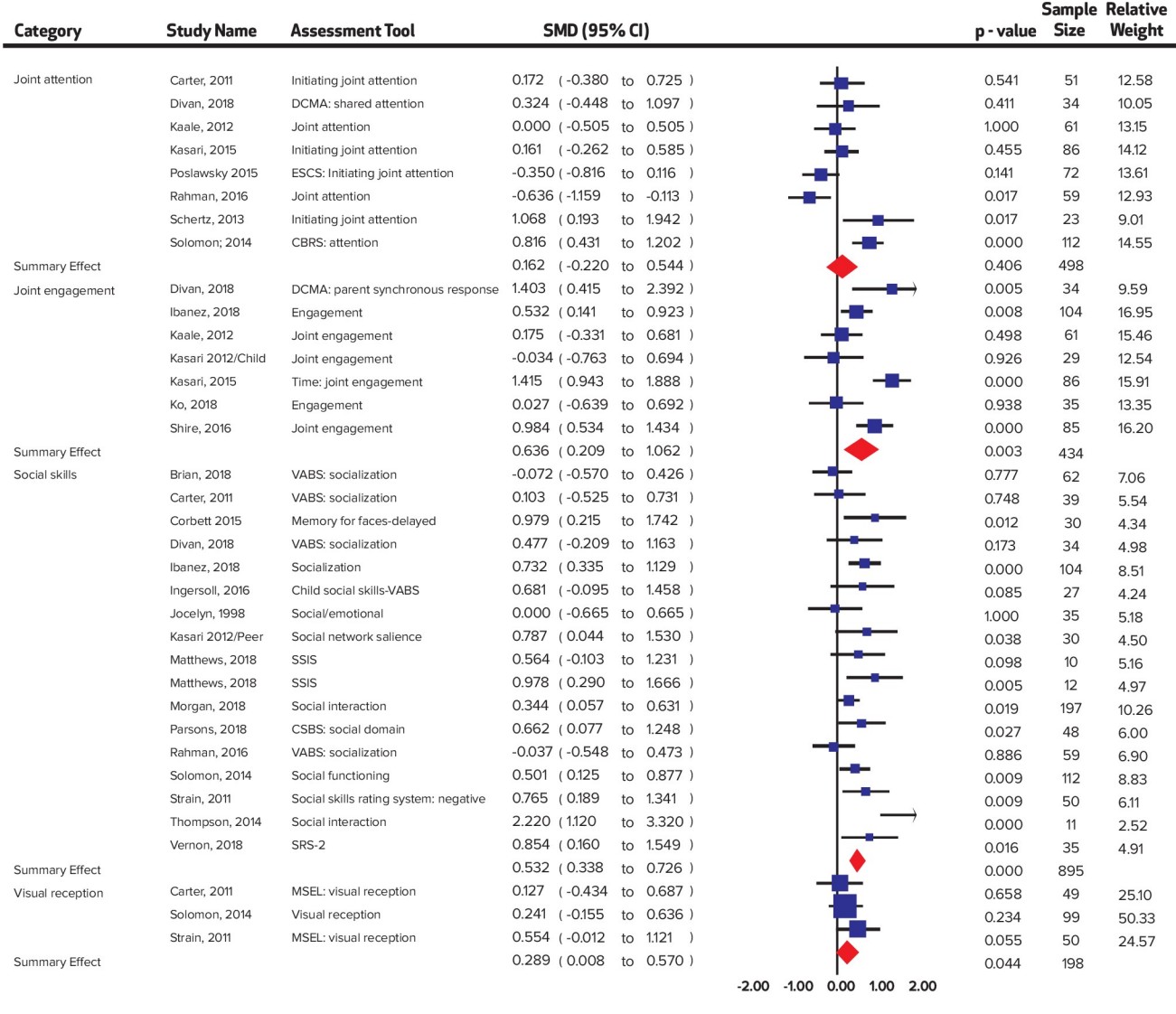

**Fig 4. Summary effect sizes for social skills.**

**Fig 5. Summary effect sizes for motor skills.**

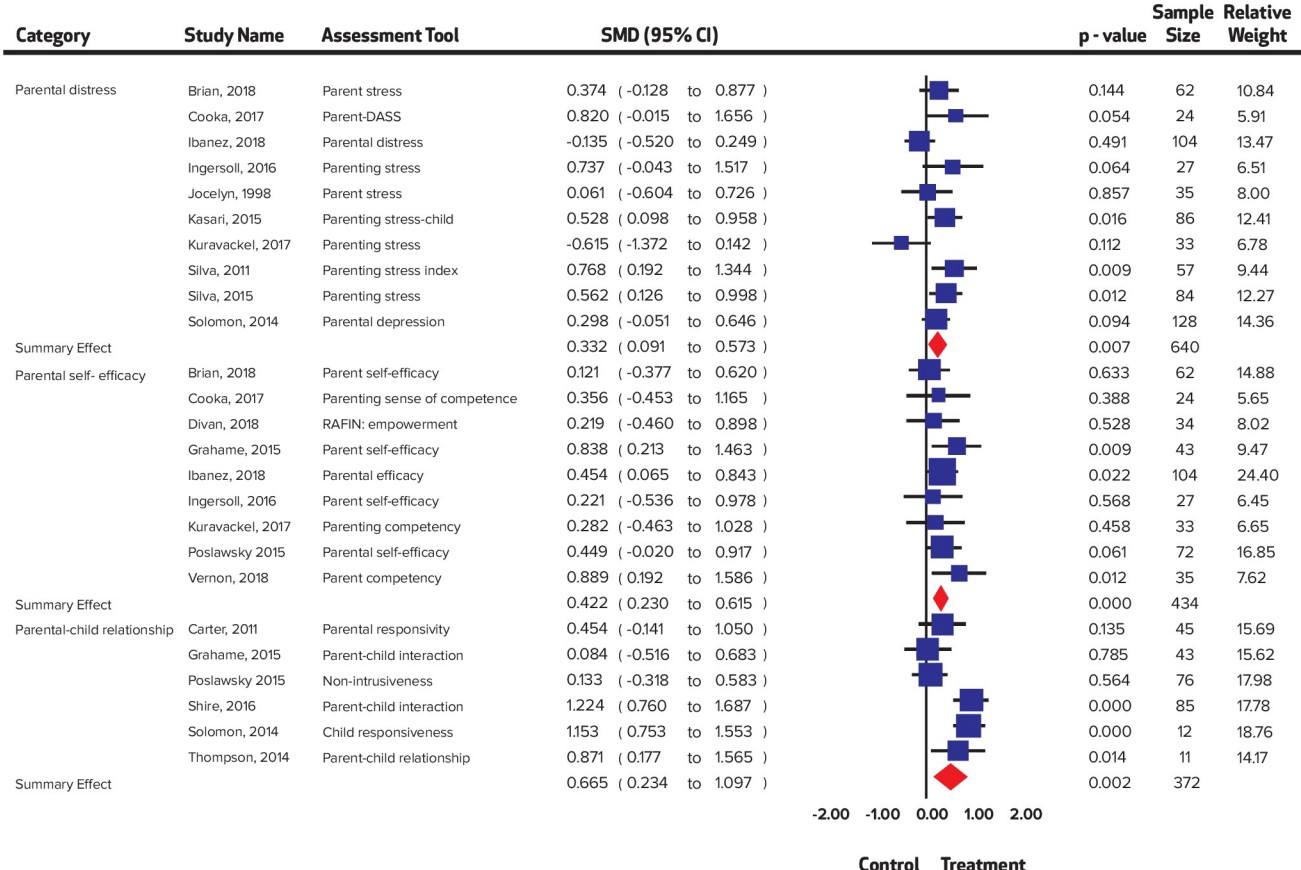

**Fig 6. Summary effect sizes for parental outcomes.**

The evidence was moderate for adaptive behaviors, severity of symptoms, receptive language, social skills, and improvement in parent child relationship. It was found to be low for joint engagement, self-efficacy and competence, and very low for joint attention (Table 4).

## Discussion

### Summary of results

We identified 33 studies comparing the effects of non-specialist mediated interventions with control groups among children with autism spectrum disorder. The meta-analyses demonstrated the effectiveness of non-specialist mediated interventions across several outcomes pertaining to social skills, motor and communication among children with autism. These were also associated with an improvement in parent-child relationship and parenting stress. The risk of bias among the studies assessed was generally low, albeit the overall strength of evidence varied across outcomes. This means that further research may change the effect estimates for a few of the outcomes reported in this review.

All of the interventions reviewed had varying density of dosage, but meta-regression analyses did not generally reveal any significant effects on the effectiveness of interventions. An outlier intervention program was the LEAP intervention program, run for 2 years spanning 834 sessions of naturalistic and incidental teaching among children with autism [23]. This

**Table 3. Subgroup analysis based on type of delivery agent.**

| Outcome | Delivery agent | Number of studies | Effect size (95% CI) | $I^2$ | $Tau^2$ | Q statistic | $p$ |
|---|---|---|---|---|---|---|---|
| Social skills | Parent | 10 | 0.42 (0.17 to 0.67) | 61.36% | 0.13 | 13.42 | 0.34 |
| | Peers | 6 | 0.75 (0.39 to 1.11) | 0% | 0 | | |
| | Teachers | 2 | 0.50 (0.03 to 0.98) | 39.17% | 0.04 | | |
| Communication | Parent | 10 | 0.12 (-0.04 to 0.29) | 0% | 0 | 6.38 | 0.04 |
| | Peers | 1 | 0.86 (0.08 o 1.63) | 0% | 0 | | |
| | Teachers | 2 | 0.46 (0.16 to 0.75) | 72.92% | 0.15 | | |
| Expressive language | Parent | 5 | 0.45 (0.03 to 0.88) | 57.66% | 0.13 | 0.08 | 0.78 |
| | Peers | 0 | - | - | | | |
| | Teachers | 1 | 0.32 (-0.52 to 1.15) | 0% | 0 | | |
| Receptive language | Parent | | 0.12 (-0.19 to 0.43) | | | 0 | 1.0 |
| | Peers | 0 | | | | | |
| | Teachers | 0 | | | | | |
| Motor skills | Parents | 5 | 0.17 (-0.08 to 0.41) | 0% | 0 | 2.72 | 0.10 |
| | Teachers | 1 | 0.69 (0.12 to 1.26) | 0% | 0 | | |
| Joint engagement | Parents | 2 | 1.01 (0.61 to 1.41) | 66.46% | 0.13 | 8.40 | 0.02 |
| | Peers | 1 | -0.002 (-0.65 to 0.65) | 0% | 0 | | |
| | Teachers | 1 | 0.18 (-0.61 to 0.96) | 0% | 0 | | |
| Joint initiation | Parents | 7 | 0.19 (-0.25 to 0.63) | 79.22% | 0.27 | 0.09 | 0.76 |
| | Peers | 0 | - | - | | | |
| | Teachers | 1 | 0 (-0.14 to 1.14) | 0% | 0 | | |
| Symptom severity | Parents | 9 | 0.44 (0.26 to 0.61) | 0% | 0 | 0.003 | 0.96 |
| | Peers | 1 | 0.42 (-0.14 to 0.98) | 0% | 0 | | |
| | Teachers | 0 | - | - | | | |
| Child distress | Parents | 3 | 0.57 (0.20 to 0.94) | 0 | 0 | 0.04 | 0.85 |
| | Peers | 1 | 0.51 (0.004 to 1.02) | 0 | 0 | | |
| | Teachers | 0 | - | 0 | - | | |
| Adaptive behavior | Parents | 5 | 0.17 (-0.17 to 0.51) | 43.85% | 0.07 | 1.56 | 0.46 |
| | Peers | 1 | 0.77 (-0.13 to 1.67) | 0% | 0 | | |
| | Teachers | 1 | 0.34 (-0.25 to 0.90) | 0% | 0 | | |
| Self-regulation | Parents | 3 | 0.54 (0.06 to 1.03) | | | | |
| | Peers | - | | | | | |
| | Teachers | - | | | | | |
| Repetitive behaviours | Parents | 3 | 0.36 (0.12 to 0.60) | | | | |
| | Peers | - | | | | | |
| | Teachers | - | | | | | |
| Visual reception | Parents | 3 | 0.11 (-0.17 to 0.39) | 0% | 0 | 1.90 | 0.17 |
| | Peers | 0 | - | | | | |
| | Teachers | 1 | 0.55 (-0.01 to 1.12) | 0% | 0 | | |
| Parental distress | Parents | 10 | 0.33 (0.09 to 0.57) | | | | |
| | Peers | 0 | | | | | |
| | Teachers | 0 | | | | | |
| Parental self-efficacy | Parents | 8 | 0.38 (0.18 to 0.58) | 0% | 0 | 1.90 | 0.17 |
| | Peers | 1 | 0.89 (0.19 to 1.59) | 0% | 0 | | |
| | Teachers | 0 | - | 0% | 0 | | |
| Parent-child relationship | Parents | 6 | 0.67 (0.23 to 1.10) | | | | |
| | Peers | 0 | - | | | | |
| | Teachers | 0 | - | | | | |

**Table 4. GRADE table for outcomes included in the systematic review.**

| № of studies | Study design | Certainty assessment | | | | | № of patients | | Effect | | Certainty | Importance |
|---|---|---|---|---|---|---|---|---|---|---|---|---|
| | | Risk of bias | Inconsistency | Indirectness | Imprecision | Other considerations | [intervention] | [comparison] | Relative (95% CI) | Absolute (95% CI) | | |
| Adaptive behaviors | | | | | | | | | | | | |
| 7 | randomised trials | not serious | not serious | not serious | Serious [a] | none | 268 | 215 | - | SMD **0.26 SD higher** (−0.001 lower to 0.52 higher) | ⊕⊕⊕◯ MODERATE | CRITICAL |
| Severity of symptoms | | | | | | | | | | | | |
| 10 | randomised trials | not serious | not serious | not serious | not serious | publication bias strongly suspected [c] | 322 | 295 | - | SMD **0.44 SD higher** (0.27 higher to 0.60 higher) | ⊕⊕⊕◯ MODERATE | CRITICAL |
| Social skills | | | | | | | | | | | | |
| 18 | randomised trials | not serious | not serious | not serious | not serious | publication bias strongly suspected [c] | 493 | 465 | - | SMD **0.52 SD higher** (0.34 higher to 0.71 higher) | ⊕⊕⊕◯ MODERATE | CRITICAL |
| Communication skills | | | | | | | | | | | | |
| 13 | randomised trials | not serious | not serious | not serious | not serious | none | 503 | 447 | - | SMD **0.23 SD higher** (0.06 higher to 0.40 higher) | ⊕⊕⊕⊕ HIGH | CRITICAL |
| Expressive language | | | | | | | | | | | | |
| 6 | randomised trials | Not serious | not serious | not serious | not serious | none | 147 | 146 | - | SMD **0.42 SD higher** (0.09 higher to 0.75 higher) | ⊕⊕⊕⊕ HIGH | CRITICAL |
| Receptive language | | | | | | | | | | | | |
| 5 | randomised trials | Not serious | Not serious | Not serious | Serious [a] | none | 151 | 163 | - | SMD **0.12 SD higher** (−0.19 lower to 0.43 higher) | ⊕⊕⊕◯ MODERATE | CRITICAL |
| Motor skills | | | | | | | | | | | | |
| 6 | randomised trials | not serious | not serious | not serious | not serious | none | 178 | 174 | - | SMD **0.21 SD higher** (−0.006 higher to 0.42 higher) | ⊕⊕⊕⊕ HIGH | CRITICAL |
| Joint attention | | | | | | | | | | | | |

*(Continued)*

**Table 4.** (Continued)

| № of studies | Study design | Risk of bias | Inconsistency | Indirectness | Imprecision | Other considerations | № of patients [intervention] | [comparison] | Relative (95% CI) | Absolute (95% CI) | Certainty | Importance |
|---|---|---|---|---|---|---|---|---|---|---|---|---|
| 8 | randomised trials | not serious | serious[b] | not serious | very serious[a] | none | 255 | 243 | - | SMD **0.16 SD higher** (0.22 lower to 0.54 higher) | ⊕○○○ VERY LOW | CRITICAL |
| **Joint engagement** | | | | | | | | | | | | |
| 7 | randomised trials | not serious | very serious[b] | not serious | not serious | none | 217 | 217 | - | SMD **0.64 SD higher** (0.21 higher to 1.06 higher) | ⊕⊕○○ LOW | CRITICAL |
| **Repetitive behaviors** | | | | | | | | | | | | |
| 3 | randomised trials | not serious | not serious | not serious | not serious | none | 130 | 127 | - | SMD **0.36 SD higher** (0.11 higher to 0.60 higher) | ⊕⊕⊕⊕ HIGH | CRITICAL |
| **Self-regulation** | | | | | | | | | | | | |
| 3 | randomised trials | not serious | serious[b] | not serious | serious[a] | none | 96 | 79 | - | SMD **0.544 SD higher** (0.06 higher to 1.028 higher) | ⊕⊕○○ LOW | CRITICAL |
| **Parental distress** | | | | | | | | | | | | |
| 10 | randomised trials | not serious | not serious | not serious | not serious | none | 334 | 306 | - | SMD **0.33 SD higher** (0.09 higher to 0.57 higher) | ⊕⊕⊕⊕ HIGH | IMPORTANT |
| **Parent-child relationship** | | | | | | | | | | | | |
| 6 | randomised trials | not serious | serious[b] | not serious | not serious | none | 199 | 183 | - | SMD **0.67 SD higher** (0.23 higher to 1.1 higher) | ⊕⊕⊕○ MODERATE | IMPORTANT |

CI: Confidence interval; SMD: Standardised mean difference

Explanations

[a] . Wide confidence intervals

[b] . Substantial heterogeneity partly explained by differences in content and delivery of interventions.

[c] . Visualization of funnel plot revealed significant publication bias

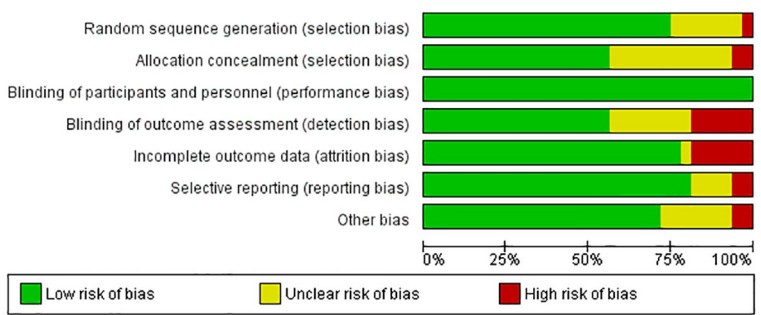

**Fig 7. Risk of bias summary.**

intervention reported the highest effect sizes, hence, we opine that sustained and long term interventions might achieve long sustaining results.

A total of three outcomes including joint engagement, parent child relationship and joint attention exhibited substantial heterogeneity ($I^2>70\%$). Rest of the outcomes presented no to moderate heterogeneity. We opine that this may be because of two main reasons. For the outcome of joint engagement, this substantial heterogeneity is due to differences in intervention content as well as different delivery agents as shown in subgroup analysis (Table 3). The outcomes of parent child relationship and joint attention were only reported in parent mediated interventions. The heterogeneity in these outcomes may be accounted for by use of different rating scale or methods of measurement. The studies reporting these outcomes used varying methods for measurement of both the joint attention and parent child relationship.

## Recommendation for task shifting

Autism spectrum disorder is a major global health concern accounting for a large disease burden, health loss, disability adjusted life years (DALYs) and high specialist treatment costs. Moreover, the scarce availability of psychiatrists and psychologists in low- and middle-income countries further aggravate this public health issue. The WHO report on global mental health infrastructure, estimated the number of psychiatrists at less than one and 7.7 mental health nurses for 100,000 people in countries inhabited by 45% of the world's population [2,43]. In 2009, there was only one registered occupational therapist in Pakistan, highlighting the bleak situation in poorly resourced countries [44].

The proven effectiveness of non-specialized autism care in present review is thus, of particular relevance to low resourced settings, where access to specialist mental health interventionists specializing in autism treatment is poor [45]. However, only one good quality randomized controlled trial conducted jointly in India & Pakistan limits the evidence for clinical and cost-effectiveness of these interventions in the region [6]. One of these studies was an adapted version of the PACT trial developed in Manchester [35] and was tested in a multi-site study conducted in Rawalpindi, Pakistan and Goa, India [6]. While, the second study revised this intervention and added a "plus" module pertaining to psychoeducation and assessment of the most disruptive comorbidity for the family [31]. Therefore, more research is required to ascertain the suitability of these interventions in the context of low- and middle-income countries. Moreover, there are no frameworks for recruitment, role descriptions and financial compensation for non-specialists, which creates a barrier in scale up and sustainability of these interventions [5, 6]. Based on the findings of this systematic review, we cannot recommend one non-specialist mediated therapy for autism. PACT, PASS and PASS plus; JASPER, SENSE and Hanen's more than words were tested in at least two studies and settings. Therefore, we

recommend that future investigators, implementors and policy makers consult these therapy programs for development of interventions suitable for their settings.

## Strengths and limitations

This study has several strengths. Firstly, the inclusion of randomized controlled studies ensured the internal validity of results. Previously, review studies had reported evidence for single cases, non-randomized controlled trials, specialist interventions or homogenous interventions based on specific strategies only [7,8,46–51]. Lastly, our study was inclusive of children of all ages, confirmed diagnoses of autism spectrum disorder, psychosocial functioning, languages and time period, thereby, improving the generalizability of the results to this study population. Lastly, the subgroup analysis based on intervention mediators led to meaningful subgroup analyses.

Despite of its strengths, there are several limitations of this review and therefore, these results should be interpreted with caution. For instance, none of the studies had reported standardized outcomes pertaining to the IQ and psychological functioning of the study sample. Meta-regression analysis accounting for IQ of the children is a necessary analysis for studying moderating effects on the intervention effects. The interventions differed in their content and strategies, study settings, and intervention mediators, leading to substantial methodological heterogeneity in the meta-analyses. Several diagnostic methods such as ADOS and different updates of DSM criteria for diagnoses of autism were employed in included RCTs, further adding heterogeneity in the results. Psychological interventions limit the blinding status of participants and personnel as well as outcome assessors that is a serious limitation. The present systematic review was based on searching of a limited number of databases, we encourage investigators to search more databases in future studies. Moreover, investigators should also consider using a more comprehensive search strategy encompassing different terms for RCT. Combining results from diverse measures applied for heterogeneous study samples is another limitation of this systematic review.

## Implications for practice and future research

The present review provides an overall good quality evidence of effectiveness of non-specialist mediated interventions among the children with autism. Most of the studies were mediated by parents and caregivers and presented low risk of bias. However, the evidence for peer and teacher mediated interventions was poor due to a limited number of studies. The sample size was low among individual studies and only a few interventions were tested in long term follow-up studies. The economic feasibility and cost-effectiveness were not reported in most of the interventions; an important metric for evaluating their suitability for task shifting and scaling up. The standardized instruments differed in studies, adding to the methodological heterogeneity among studies. Future studies should be designed keeping these limitations in context, emphasizing the introduction of standardized and cross-culturally validated instruments for assessment of symptomatology.

## Recommendations

Despite of the aforementioned limitations, a small to moderate improvements in several debilitating symptoms of autism were noticed. These interventions also reduced care-giver stress and improved parent-child relationship. Based on the clinical effectiveness and good quality of evidence for these interventions, we recommend up-scaling of these interventions in high income countries. However, more research is required to ascertain the suitability of these interventions in context of low- and middle-income countries. Based on the findings of this

systematic review, we cannot recommend one non-specialist mediated therapy for autism. PACT, PASS and PASS plus; JASPER, SENSE and Hanen's more than words were tested in at least two studies and settings.

## Supporting information

**S1 File. PRISMA checklist.**
(DOC)

**S2 File. Search strategy.** Search strategy for all databases utilized in this study.
(XLSX)

**S3 File. Dataset for meta-analysis.** Data set used for meta-analysis in CMA format.
(CMA)

**S4 File. Supplementary figures in the manuscript.** This file has following figures: a) Meta-regression analysis for quality of studies b) Meta-regression analysis for duration of intervention program c) Meta-regression analysis for number of sessions of intervention program d) Funnel plot for social skills e) Funnel plot for severity of symptoms f) Risk of bias for all studies.
(DOCX)

## Author Contributions

**Conceptualization:** Sadiq Naveed, Ahmed Waqas, Sherief Ghozy, Anum Saeed Khan, Atif Rahman.

**Data curation:** Sadiq Naveed, Ahmed Waqas, Afshan Naz Amray, Raheel Imtiaz Memon, Nisma Javed, Muhammad Annas Tahir, Sherief Ghozy, Nusrat Jahan.

**Formal analysis:** Sadiq Naveed, Ahmed Waqas, Atif Rahman.

**Investigation:** Ahmed Waqas.

**Methodology:** Ahmed Waqas.

**Project administration:** Ahmed Waqas.

**Resources:** Ahmed Waqas.

**Software:** Ahmed Waqas.

**Supervision:** Ahmed Waqas.

**Validation:** Ahmed Waqas.

**Visualization:** Ahmed Waqas.

**Writing – original draft:** Sadiq Naveed, Ahmed Waqas, Afshan Naz Amray, Raheel Imtiaz Memon, Nisma Javed, Muhammad Annas Tahir, Sherief Ghozy, Nusrat Jahan, Anum Saeed Khan.

**Writing – review & editing:** Sadiq Naveed, Ahmed Waqas, Afshan Naz Amray, Raheel Imtiaz Memon, Nisma Javed, Atif Rahman.

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
