## [Decision Letter · Decision Letter 0]

28 Aug 2019

PONE-D-19-19450

Implementation and effectiveness of non-specialist mediated interventions for children with Autism

Spectrum Disorder: A systematic review and meta-analysis

PLOS ONE

Dear Ahmed Waqas,

Thank you for submitting your manuscript to PLOS ONE. After careful consideration, we feel that it has merit but does not fully meet PLOS ONE’s publication criteria as it currently stands. Therefore, we invite you to submit a revised version of the manuscript that addresses the points raised during the review process.

ACADEMIC EDITOR:  The reviewers have raised a number of points which we believe major modifications are necessary to improve the manuscript, taking into account the reviewers' remarks.  Please consider and address each of the comments raised by the reviewers before resubmitting the manuscript. This letter should not be construed as implying acceptance, as a revised version will be subject to re-review.

We would appreciate receiving your revised manuscript by Oct 12 2019 11:59PM. To enhance the reproducibility of your results, we recommend that if applicable you deposit your laboratory protocols in protocols.io, where a protocol can be assigned its own identifier (DOI) such that it can be cited independently in the future. For instructions see: http://journals.plos.org/plosone/s/submission-guidelines#loc-laboratory-protocols

We look forward to receiving your revised manuscript.

Kind regards,

Wisit Cheungpasitporn, MD, FACP

University of Mississippi Medical Center

Twitter: @wisit661 Email: wcheungpasitporn@gmail.com 

Academic Editor

PLOS ONE

Journal Requirements:

3. Please upload a copy of Figure 4, to which you refer in your text on page 11. If the figure is no longer to be included as part of the submission please remove all reference to it within the text

Reviewers' comments:

Reviewer's Responses to Questions

**Comments to the Author**

1. Is the manuscript technically sound, and do the data support the conclusions?

Reviewer #1: Partly

Reviewer #2: Yes

Reviewer #3: Yes

Reviewer #4: Yes

Reviewer #5: Partly

2. Has the statistical analysis been performed appropriately and rigorously? 

Reviewer #1: No

Reviewer #2: Yes

Reviewer #3: Yes

Reviewer #4: I Don't Know

Reviewer #5: Yes

3. Have the authors made all data underlying the findings in their manuscript fully available?

Reviewer #1: Yes

Reviewer #2: Yes

Reviewer #3: Yes

Reviewer #4: Yes

Reviewer #5: Yes

4. Is the manuscript presented in an intelligible fashion and written in standard English?

Reviewer #1: No

Reviewer #2: Yes

Reviewer #3: Yes

Reviewer #4: Yes

Reviewer #5: No

5. Review Comments to the Author

Reviewer #1: This is an important and interesting paper, but there are numerous grammatical errors, and word omissions. Given the sophistication of your approach and the immense effort involved, it is curious that you chose to use the number of participants as a weighting scheme. Why did you not choose inverse variance or the Hedges & Olkin or Hunter & Schmidt estimators of optimal weights?

When you say that, "Heterogeneity was considered significant at a cut off value of GE 40%," does that mean that you used a fixed-effect approach for those under 40% and random-effects for those greater than or equal to 40%? This should be clarified in the manuscript.

I suspect that a random-effects approach throughout the analysis would yield more realistic estimates of effect size given the differences across studies in measurement instruments, procedures, treatments, and participants.

Reviewer #2: Thank you for the opportunity to review this interesting manuscript. Its subject is an important topic and it makes a helpful contribution to the literature. Although well-written in many parts, it would benefit from a careful edit of the English in places to maximise comprehensibility, and I highlight some issues below which should be clarified before it can be accepted for publication.

ABSTRACT

The final sentence of the Introduction section seems to be more of a conclusion.

It is unclear what 'academic search' means. Specific databases should usually be mentioned in the abstract. On which date was the search completed? Were only randomised controlled trials included? Were self-report and objective assessments of outcomes combined?

The English of this section could be tweaked to enhance its clarity.

INTRODUCTION

This section is clear and well-written.

METHODS

A specific search date would be helpful. The abstract states 'through 2018' but the methods say 'through January 2019'. Outside North America, the term 'through' may not be widely understood in this sense.

This is a relatively truncated set of databases to search. Is there a reason, given the authors' interest in LMICs, that other databases such as Global Health or Embase were not included?

The search terms to specify RCTs are quite limited, not including the word 'randomised'. Can the authors be sure that this strategy did not miss eligible studies? The fact that only 659 non-duplicate results were obtained from the search does suggest that this search strategy was extremely narrow.

The title specifies interventions for children but the Inclusion criteria state that no age restrictions were applied, suggesting that studies of adults were included.

What is meant by 'unreliably extracted' data?

Does 'overlapped data sets' mean papers reporting the same study? This could be clearer.

Data analysis: how similar were outcomes required to be in order to be meta-analysed? Were self-report, parent-report, teacher-reported etc. measures combined?

RESULTS

Please define CAMH.

Density of dosage: is (158.07) intended to indicate the standard deviation? This should be stated.

Did two studies genuinely report sessions lasting 1500 minutes, i.e. 25 hours? Can this really have been continuous? Might such an intervention have caused harm?

Please define (if applicable) the meanings of PACT, PASS, QST, JASPER, HANEN, COMPASS, PEERS acronyms.

It is distracting that the referencing style changes from numerical to naming authors halfway through the results.

Under outcomes, please clarify if self-report and reports by heterogeneous others were combined.

It is confusing that "a significant improvement was noted in child distress (SMD=0.23...)" and also "no significant improvement was noted in child distress (SMD=0.158...)" - is this an error?

p11: "allocation concelment (13), selective reporting (4)..." - what do these numbers mean? Also high/unclear risk of bias?

The terms "risk of bias" and quality are conflated in this paragraph which is confusing.

DISCUSSION

Did every included study use TAU as the comparator?

Please define DALYs

Some mention of the ethical and sustainability issues around non-specialists not seeming to have been paid would enhance the discussion

The authors only briefly mention that two included studies came from LMICs and that one was of good quality. A little more detail on both studies would be of interest, given the focus on task sharing for LMICs.

The statement "This would essentially reduce the treatment gap for children with autism, and ensure mental health for all across the globe" does not clearly refer to something in the previous sentence (the "this" of this sentence) and it is debatable that reducing the treatment gap for children with autism would indeed ensure MH for all across the globe.

IQ may be expected to moderate treatment effects but would it mediate them?

The authors state that cost-effectiveness was not reported in most studies but then conclude that "based on the... cost-effectiveness... we recommend up-scaling of these interventions".

Were some interventions more effective than others? Would the authors advocate up-scaling any models in particular?

FIGURES

The forest plots are very cramped and would benefit from larger sizing and greater separation. Does Total mean N? A column for N/n (intervention vs control sample size) would help.

Acronyms in the forest plot impede understanding, e.g. 'PLS' - using simple language here would help.

It might help to order the forest plots in order of strongest to weakest effect size.

The forest plot pages are divided into primary outcomes and secondary outcomes (although the latter is not labelled) but these distinctions are not clearly drawn in the manuscript. Doing so would be helpful, for clarification and to align the figures with the text.

The risk of bias figure requires a heading and its meaning is not clear. Do percentages refer to the percentage of included papers graded as low, unclear or high risk? It would be more informative to show a table of the included studies and their individual gradings in each column.

Reviewer #3: Dear authors,

I appreciate your manuscript and I do not have any substantial comments on it.

Best regards

The reviewer

Reviewer #4: The manuscript in my opinion is technically sound.

I am not very competent in statistics to comment on the accuracy. A statistical expert may comment.

All data is available for review.

The English is standard and intelligible. The language used is simple for the reader’s understanding. The flow of the text is also fine in my opinion.

Additionally, the topic chosen is relevant as it highlights the difference between the highly technical and the basic non pharmacological interventions in ASD in the backdrop of the cost differences between the two. The study being a meta analysis involving 33 studies is a strength. The limitations are also highlighted giving the reader an idea of possible biases.

Reviewer #5: This meta-analysis has not been registered online. Please add this point in the limitation.

Literature Searches and Search terms are incomplete. This is suboptimal for publication for systematic review. Search terms in each database are different. Please attach search terms that were used in each database as supplement for Data source and search strategies in the manuscript. Please provide details search terms in supplementary documents. Please attach syntax used in each database as supplementary.

Please also include timeline of the literature search in the method section of the abstract.

It will be better to show kappa for the selection and data extraction. Please show the data of kappa of agreement during the systematic searches. How disagreements were solved during the systematic search among independent reviewers?

There is still a considerable heterogeneity as in your limitation. Meta-regression analysis is then strongly recommended.

There is substantive heterogeneity in some outcomes. It also is unclear whether the t-statistic is being used for the degrees of freedom in the random effects analysis (i.e., N-1 d.f. not asymptotic [1.96] value multiplied by tau). Please assure that the t-statistic (or Satterthwaite correction) is being used and add that information to the Methods, when the number of studies is small (e.g., < 10). Apply this principle throughout the author's paper. For reference, the authors can refer the article “IntHout J, Ioannidis JP, Borm GF. The Hartung-Knapp-Sidik-Jonkman method for random effects meta-analysis is straightforward and considerably outperforms the standard DerSimonian-Laird method. BMC Medical Research Methodology 2014;14:25.” The issue is the Student t statistic.

Authors should discuss the reason of heterogeneity.

Please make the data for this review publicly available, possibly through the Open Science Framework (osf.io). Items to include: list of excluded studies, commands forstatistical analysis, spreadsheets or data used for the meta-analyses, etc. Making data publicly available will promote the reproducibility of the review and is best practices for systematic reviews and meta-analyses.

Some revision of the English language is needed. There are some parts of the paper where it is quite difficult to make sense of some sentences. English edit will help to improve the quality of the manuscript.

6. PLOS authors have the option to publish the peer review history of their article (what does this mean?). If published, this will include your full peer review and any attached files.

Reviewer #1: No

Reviewer #2: Yes: Dr R Keynejad

Reviewer #3: No

Reviewer #4: No

Reviewer #5: No

---

## [Author Response · Author response to Decision Letter 0]

13 Sep 2019

Dear Professor Cheungpasitporn,

 We are very grateful to you and the reviewers for such an excellent feedback. This has substantially improved the quality of the manuscript. We have revised our manuscript in line with your suggestions and provide point by point responses to the reviewers below.

Please, do not hesitate to contact us if further revisions are needed. We hope for a favorable decision in due time. 

Best wishes,

Dr. Ahmed Waqas

Corresponding author

Reviewer #1

Comment 1:

This is an important and interesting paper, but there are numerous grammatical errors, and word omissions. Given the sophistication of your approach and the immense effort involved, it is curious that you chose to use the number of participants as a weighting scheme. Why did you not choose inverse variance or the Hedges & Olkin or Hunter & Schmidt estimators of optimal weights?

Response

My co-authors & I are very grateful to you for your excellent feedback on the manuscript. We have now thoroughly proof read the paper for grammatical errors and accounted for word omissions in the manuscript.

We have now provided forest plots along with relative weights (random effects) assigned to each study. We had used inverse variance method and this has now been mentioned in the manuscript as well.

Comment 2:

When you say that, "Heterogeneity was considered significant at a cut off value of GE 40%," does that mean that you used a fixed-effect approach for those under 40% and random-effects for those greater than or equal to 40%? This should be clarified in the manuscript. I suspect that a random-effects approach throughout the analysis would yield more realistic estimates of effect size given the differences across studies in measurement instruments, procedures, treatments, and participants.

Response

Thank you for a thoughtful comment. We have now revised this statement which reads as, “Depending on the extent of heterogeneity, data were pooled together using either the fixed or random effects. Heterogeneity was considered significant at a cut off value > 40% in which case random effects analysis was used [17].” 

We only employed random effects analysis when I2 > 40%- and I believe our criteria to be more firm when compared with recommendations in Cochrane handbook by Higgins and Green (2011). Moreover, most of the outcomes presented significant heterogeneity and therefore were analyzed using random effects. 

Reviewer #2

General feedback

Thank you for the opportunity to review this interesting manuscript. Its subject is an important topic and it makes a helpful contribution to the literature. Although well-written in many parts, it would benefit from a careful edit of the English in places to maximise comprehensibility, and I highlight some issues below which should be clarified before it can be accepted for publication.

Response

Dear Sir/Madam, we are very grateful to you for providing such a thorough feedback on our manuscript. We have carefully proof read and edited the manuscript to avoid any mistakes or errors in use of English. We believe it has significantly improved the comprehensibility and readability of the manuscript. 

Comment 1

ABSTRACT

The final sentence of the Introduction section seems to be more of a conclusion.

It is unclear what 'academic search' means. Specific databases should usually be mentioned in the abstract. On which date was the search completed? Were only randomised controlled trials included? Were self-report and objective assessments of outcomes combined?

The English of this section could be tweaked to enhance its clarity.

Response

We have rephrased the first sentence as, “In recent years, several non-specialist mediated interventions have been developed and tested to address problematic symptoms associated with autism. These can be implemented”.

Academic search has been rephrased to electronic search. We have also mentioned dates, inclusion of RCTs and use of objective outcomes. 

These sentences now reads as, “An electronic search was conducted in eight academic databases since their inception to 31st December 2018…………”.

“A total of 31 randomized controlled trials were published post-2010 while only 2 were published prior to it”.

We have thoroughly proof read the abstract to improve the use of English in it. 

Comment 2

INTRODUCTION

This section is clear and well-written.

Response

We are very grateful to you for your kind feedback.

Comment 3

METHODS

a) A specific search date would be helpful. The abstract states 'through 2018' but the methods say 'through January 2019'. Outside North America, the term 'through' may not be widely understood in this sense.

b) This is a relatively truncated set of databases to search. Is there a reason, given the authors' interest in LMICs, that other databases such as Global Health or Embase were not included?

c) The search terms to specify RCTs are quite limited, not including the word 'randomised'. Can the authors be sure that this strategy did not miss eligible studies? The fact that only 659 non-duplicate results were obtained from the search does suggest that this search strategy was extremely narrow.

d) The title specifies interventions for children but the Inclusion criteria state that no age restrictions were applied, suggesting that studies of adults were included.

e) What is meant by 'unreliably extracted' data?

f) Does 'overlapped data sets' mean papers reporting the same study? This could be clearer.

g) Data analysis: how similar were outcomes required to be in order to be meta-analysed? Were self-report, parent-report, teacher-reported etc. measures combined?

Response

a) We have rephrased this sentence which reads as, “An academic search was conducted in eight electronic databases including PubMed, Scopus, Web of Science, POPLINE, New York Academy of Medicine, PsycINFO, Psycharticles, and CINAHL, from their inception to 31st December 2018. , using following search terms:”

b) Dear Sir/Madam, this is a very valid comment. We searched eight electronic databases out of which three were major ones. This is in accordance with AMSTAR checklist for appraisal of systematic reviews which recommends atleast 2 major database searches and one grey literature search (Reference: https://amstar.ca/docs/AMSTARguideline.pdf). 

Pubmed has one of the largest coverage for medical journals, whereas scopus and web of science both cover medical, social and multidisciplinary journals. Moreover, other minor databases including CINAHL and psychinfo are specialized databases for nursing and psychiatry. And popline and NYAM cover grey literature. 

Web of science database also searches other databases such as medline, Scielo and Korean journal databases by default (Reference: https://clarivate.libguides.com/webofscienceplatform/coverage.) 

Citation Indexes in Web of Science:

Web of Science Core Collection

BIOSIS Citation Index

Chinese Science Citation Database

Data Citation Index

Russian Science Citation Index

SciELO Citation Index

Subject specialized and regional indexes:

Biological Abstracts, BIOSIS Previews

CABI: CAB Abstracts and Global Health

FSTA—the food science resource

Inspec

KCI—Korean Journal Database

Medline

Zoological Record

c) Although the term randomized or controlled was not used, search term trial should be sensitive enough to pick most of the articles. Adding the terms, randomized would have decreased the number of studies by making them more specific. 

d) The statement, No restrictions or database filters regarding language, time period or publication year were applied reflects that we did not use any limiters or filters in the databases. We have now mentioned “database filters” to make it more precise.

e) The terms “unreliably extracted data” has been deleted.

f) Yes, it does! It has now been rephrased to “Overlapping data sets reporting results from same study.”

g) We have now included more information for selection of autism and also provide an example for it. “For outcomes, an apriori decision was taken to include all types of psychometric testing whether conducted by specialists, teachers or parents. A variety of psychometric scales used for measurement of symptoms of autism are reported in the literature. We conducted a thorough audit of included studies to identify the psychometric scales used and categorized them under a unifying category. For instance, total symptom severity comprised of several scales such as Autism Diagnostic Observation Schedule; Autism Behaviour Checklist; Vineland Adaptive Behaviour Scale and Childhood Autism Rating Scale among others.”

Comment 4

RESULTS

a) Please define CAMH.

b) Density of dosage: is (158.07) intended to indicate the standard deviation? This should be stated.

c) Did two studies genuinely report sessions lasting 1500 minutes, i.e. 25 hours? Can this really have been continuous? Might such an intervention have caused harm?

d) Please define (if applicable) the meanings of PACT, PASS, QST, JASPER, HANEN, COMPASS, PEERS acronyms.

e) It is distracting that the referencing style changes from numerical to naming authors halfway through the results.

f) Under outcomes, please clarify if self-report and reports by heterogeneous others were combined.

g) It is confusing that "a significant improvement was noted in child distress (SMD=0.23...)" and also "no significant improvement was noted in child distress (SMD=0.158...)" - is this an error?

h) p11: "allocation concelment (13), selective reporting (4)..." - what do these numbers mean? Also high/unclear risk of bias?

i) The terms "risk of bias" and quality are conflated in this paragraph which is confusing.

Response

a) CAMH has been defined as “multidisciplinary child and adolescent mental health”.

b) Yes, it was standard deviation and has now reported. 

c) Only one study mentioned implementation of SCERTS intervention (Morgan, 2016) lasting 1 week for 25 hours (1500 minutes). I have now specified this as “weekly”.

d) All of these abbreviation have now been presented in full form.

e) Numerical references have now been provided for this section. 

f) We have included a new statement clarifying this: “For the purpose of meta-analysis, we combined effect sizes on all types of outcomes reported by teachers, parents or experts.”

g) This was indeed an error and has been removed from the revised manuscript. 

h) We have now rephrased this sentence to “Frequency of studies reporting a high risk across other domains of Cochrane risk of bias tool were: Blinding of outcome assessors (n=14), other sources of bias (n=9), attrition bias (n=8), selective reporting (n=4) and blinding of participants and personnel (n=0). A total of 11 studies were rated as having low quality > 3 matrices rated as having unclear or high risk of bias [22,23,26,27,32,41] (Figure 4 and Supplementary Figure 6).” Figure 4 presents a clustered bar chart exhibiting frequencies of high, unclear and low risk bias across all matrices of Cochrane risk of bias tool. Supplementary figure 6 presents study wise risk of bias across all matrices of Cochrane risk of bias tool.

i) This has now been rectified. The term “Risk of bias” has now been used throughout the manuscript. 

Comments

DISCUSSION

a) Did every included study use TAU as the comparator?

b) Please define DALYs

c) Some mention of the ethical and sustainability issues around non-specialists not seeming to have been paid would enhance the discussion

d) The authors only briefly mention that two included studies came from LMICs and that one was of good quality. A little more detail on both studies would be of interest, given the focus on task sharing for LMICs.

e) The statement "This would essentially reduce the treatment gap for children with autism, and ensure mental health for all across the globe" does not clearly refer to something in the previous sentence (the "this" of this sentence) and it is debatable that reducing the treatment gap for children with autism would indeed ensure MH for all across the globe.

f) IQ may be expected to moderate treatment effects but would it mediate them?

g) The authors state that cost-effectiveness was not reported in most studies but then conclude that "based on the... cost-effectiveness... we recommend up-scaling of these interventions".

h) Were some interventions more effective than others? Would the authors advocate up-scaling any models in particular?

Responses

a) TAU has been replaced with control group.

b) DALYs full form “disability adjusted life years” has now been provided. 

c) Following sentence has been added, “Moreover, there are no frameworks for recruitment, role descriptions and financial compensation for non-specialists, which creates a barrier in scale up and sustainability of these interventions [5, 6].”

d) We have now added more information pertaining to these studies: “However, only one good quality randomized controlled trial conducted jointly in India & Pakistan limits the evidence for clinical and cost-effectiveness of these interventions in the region [6]. One of these studies was an adapted version of the PACT trial developed in Manchester [35] and was tested in a multi-site study conducted in Rawalpindi, Pakistan and Goa, India [6]. While, the second study revised this intervention and added a “plus” module pertaining to psycho-education and assessment of the most disruptive comorbidity for the family [31].”

e) This sentence has been revised to “Access to non-specialists would essentially reduce the treatment gap for children with autism, and ensure mental health for all across the globe [45].”

f) The term mediating has now been deleted.

g) The term cost-effectiveness has now been deleted. 

h) Based on the findings of this systematic review, we cannot recommend one non-specialist mediated therapy for autism. PACT, PASS and PASS plus; JASPER, SENSE and Hanen’s more than words were tested in at least two studies and settings.

Comments

FIGURES

The forest plots are very cramped and would benefit from larger sizing and greater separation. Does Total mean N? A column for N/n (intervention vs control sample size) would help.

Acronyms in the forest plot impede understanding, e.g. 'PLS' - using simple language here would help.

It might help to order the forest plots in order of strongest to weakest effect size.

The forest plot pages are divided into primary outcomes and secondary outcomes (although the latter is not labelled) but these distinctions are not clearly drawn in the manuscript. Doing so would be helpful, for clarification and to align the figures with the text.

The risk of bias figure requires a heading and its meaning is not clear. Do percentages refer to the percentage of included papers graded as low, unclear or high risk? It would be more informative to show a table of the included studies and their individual gradings in each column.

Response

New forest plots have now been added and classified as communication; symptoms; motor symptoms; symptom severity and parental outcomes. 

We have also added two sentence to ease the interpretation of risk of bias tool: “Figure 4 presents a clustered bar chart exhibiting frequencies of high, unclear and low risk bias across all matrices of Cochrane risk of bias tool. Supplementary figure 6 presents study wise risk of bias across all matrices of Cochrane risk of bias tool.” Individual gradings for each study are presented as supplementary figure 6.

Reviewer #3

Dear authors, I appreciate your manuscript and I do not have any substantial comments on it.

Best regards

The reviewer

Response

 Dear Sir or Madam, we are very grateful to you for your kind feedback. 

Reviewer #4

Comment

The manuscript in my opinion is technically sound.

I am not very competent in statistics to comment on the accuracy. A statistical expert may comment.

All data is available for review.

The English is standard and intelligible. The language used is simple for the reader’s understanding. The flow of the text is also fine in my opinion.

Additionally, the topic chosen is relevant as it highlights the difference between the highly technical and the basic non pharmacological interventions in ASD in the backdrop of the cost differences between the two. The study being a meta analysis involving 33 studies is a strength. The limitations are also highlighted giving the reader an idea of possible biases.

Response

Dear Sir or Madam, we are very grateful to you for your kind review of this manuscript and encouraging feedback.

Reviewer #5

Comment 1

This meta-analysis has not been registered online. Please add this point in the limitation.

Response

The manuscript has been registered online. The protocol registered in PROSPERO is numbered (CRD42017066009). Please, access it using this URL: https://www.crd.york.ac.uk/prospero/display_record.php?RecordID=66009

Comment 2

Literature Searches and Search terms are incomplete. This is suboptimal for publication for systematic review. Search terms in each database are different. Please attach search terms that were used in each database as supplement for Data source and search strategies in the manuscript. Please provide details search terms in supplementary documents. Please attach syntax used in each database as supplementary.

Response

We have now provided search terms adapted for each database and provided as a supplementary file. 

Comment 4

Please also include timeline of the literature search in the method section of the abstract.

Response

We have now provided with the timeline of literature search which now reads as, “An academic search was conducted in eight electronic databases including PubMed, Scopus, Web of Science, POPLINE, New York Academy of Medicine, PsycINFO, Psycharticles, and CINAHL, from their inception to 31st December 2018. , using following search terms:…”

Comment 5

It will be better to show kappa for the selection and data extraction. Please show the data of kappa of agreement during the systematic searches. How disagreements were solved during the systematic search among independent reviewers?

Response

Process of disagreement resolution has now been outlined in the manuscript. It reads as, “All data were extracted independently by three teams of reviewers using manualized data extraction forms and any disagreements among the reviewers, were resolved through discussion in conjunction with a senior author.” 

However, we did not calculate cohen’s kappa in this case. According to our understanding, Cohen’s kappa is necessary when extractions are done with fewer personnel. For instance, in cases where, a junior reviewer works with a senior reviewer; and the senior reviewer extracts data for a few articles (say 20% of the total) and then checks inter rater reliability with the junior reviewer. In contrast, in present study three teams of reviewers extracted the data independently, where each article was rated by two reviewers working independently from each other, and in conjunction with senior authors.

Comment 6

There is still a considerable heterogeneity as in your limitation. Meta-regression analysis is then strongly recommended.

There is substantive heterogeneity in some outcomes. It also is unclear whether the t-statistic is being used for the degrees of freedom in the random effects analysis (i.e., N-1 d.f. not asymptotic [1.96] value multiplied by tau). Please assure that the t-statistic (or Satterthwaite correction) is being used and add that information to the Methods, when the number of studies is small (e.g., < 10). Apply this principle throughout the author's paper. For reference, the authors can refer the article “IntHout J, Ioannidis JP, Borm GF. The Hartung-Knapp-Sidik-Jonkman method for random effects meta-analysis is straightforward and considerably outperforms the standard DerSimonian-Laird method. BMC Medical Research Methodology 2014;14:25.” The issue is the Student t statistic.

Response

Meta-regression was indeed applied for a number of moderators. These moderators included: age, year of publication or duration of program and session and number of sessions or quality of trials. These have been reported in the manuscript. And scatter plots for all of these variables have been provided as supplementary files. 

Moreover, we also conducted subgroup analyses for type of delivery agents (parents, teachers and peers). These have also been reported in the revised manuscript under the heading of moderator analysis. We have also added table 3 presenting subgroup analysis based on delivery agents. Regarding the issue of t-statistic, neither is it calculable in Comprehensive Meta-analysis software nor is it widely/frequently used in meta-analyses. However, we have now provided detailed statistics to account for heterogeneity. We have provided detailed subgroup analyses as well as corresponding I2, Tau2, Q-value and P-value. These are more widely used indicators of heterogeneity and may make better sense for the readers. 

Comment 7

Authors should discuss the reason of heterogeneity.

Response

We have now added a paragraph on reasons for heterogeneity in discussion section. It reads as, “A total of three outcomes including joint engagement, parent child relationship and joint attention exhibited substantial heterogeneity (I2>70%). Rest of the outcomes presented no to moderate heterogeneity. We opine that this may be because of two main reasons. For the outcome of joint engagement, this substantial heterogeneity is due to differences in intervention content as well as different delivery agents as shown in subgroup analysis (Table 3). The outcomes of parent child relationship and joint attention were only reported in parent mediated interventions. The heterogeneity in these outcomes may be accounted for by use of different rating scale or methods of measurement. The studies reporting these outcomes used varying methods for measurement of both the joint attention and parent child relationship.:

Comment 8

Please make the data for this review publicly available, possibly through the Open Science Framework (osf.io). Items to include: list of excluded studies, commands for statistical analysis, spreadsheets or data used for the meta-analyses, etc. Making data publicly available will promote the reproducibility of the review and is best practices for systematic reviews and meta-analyses.

Response

We have now provided the data requested as supplementary files. All the data related to characteristics of included studies, summary of risk of bias and detailed risk of bias for individual studies have already been given in the manuscript or as supplementary files. 

Comment 8

Some revision of the English language is needed. There are some parts of the paper where it is quite difficult to make sense of some sentences. English edit will help to improve the quality of the manuscript.

Response

We have now thoroughly proof read the revised manuscript for any language errors and omissions.

---

## [Decision Letter · Decision Letter 1]

1 Oct 2019

PONE-D-19-19450R1

Implementation and effectiveness of non-specialist mediated interventions for children with Autism Spectrum Disorder: A systematic review and meta-analysis

PLOS ONE

Dear Ahmed Waqas,

Thank you for submitting your manuscript to PLOS ONE. After careful consideration, we feel that it has merit but does not fully meet PLOS ONE’s publication criteria as it currently stands. Therefore, we invite you to submit a revised version of the manuscript that addresses the points raised during the review process.

ACADEMIC EDITOR: Our expert reviewer(s) have additionally recommended some minor revisions to your revised manuscript. Therefore, I invite you to respond to the reviewer(s)' comments as below and revise your manuscript.

We would appreciate receiving your revised manuscript by Nov 15 2019 11:59PM. To enhance the reproducibility of your results, we recommend that if applicable you deposit your laboratory protocols in protocols.io, where a protocol can be assigned its own identifier (DOI) such that it can be cited independently in the future. For instructions see: http://journals.plos.org/plosone/s/submission-guidelines#loc-laboratory-protocols

We look forward to receiving your revised manuscript.

Kind regards,

Wisit Cheungpasitporn, MD, FACP

University of Mississippi Medical Center

Twitter: @wisit661 Email: wcheungpasitporn@gmail.com 

Academic Editor

PLOS ONE

Reviewers' comments:

Reviewer's Responses to Questions

**Comments to the Author**

1. If the authors have adequately addressed your comments raised in a previous round of review and you feel that this manuscript is now acceptable for publication, you may indicate that here to bypass the “Comments to the Author” section, enter your conflict of interest statement in the “Confidential to Editor” section, and submit your "Accept" recommendation.

Reviewer #1: All comments have been addressed

Reviewer #2: (No Response)

Reviewer #3: All comments have been addressed

Reviewer #4: All comments have been addressed

Reviewer #5: All comments have been addressed

2. Is the manuscript technically sound, and do the data support the conclusions?

Reviewer #1: Yes

Reviewer #2: Yes

Reviewer #3: Yes

Reviewer #4: Yes

Reviewer #5: Yes

3. Has the statistical analysis been performed appropriately and rigorously? 

Reviewer #1: Yes

Reviewer #2: Yes

Reviewer #3: Yes

Reviewer #4: I Don't Know

Reviewer #5: Yes

4. Have the authors made all data underlying the findings in their manuscript fully available?

Reviewer #1: Yes

Reviewer #2: Yes

Reviewer #3: Yes

Reviewer #4: Yes

Reviewer #5: Yes

5. Is the manuscript presented in an intelligible fashion and written in standard English?

Reviewer #1: Yes

Reviewer #2: Yes

Reviewer #3: Yes

Reviewer #4: Yes

Reviewer #5: (No Response)

6. Review Comments to the Author

Reviewer #1: All of my comments were addressed. I have no further comments.

Reviewer #2: Thank you for your responses to my previous review; the changes made have enhanced the manuscript. Please see some comments below which should be addressed before acceptance for publication.

I agree with Reviewer #1 that random-effects meta-analyses across the publication as a whole would be preferable and clearer to the reader. Would the authors consider following this suggestion?

Methods Comment c) When I suggested including the search term 'randomized' I intended this to refer to an alternative to RCT rather than an additional requirement, so this would not have decreased the number of results. The authors have not addressed my question about the very narrow search. I would suggest that the narrowness of the search and the number of databases be included in the discussion as limitations.

Methods Comment d) The authors have not clarified my query about whether the results represent a study of any interventions for participants of any age or whether in fact only studies of children were included. This needs to be clear throughout the manuscript.

Methods Comment g) I would suggest that combining results from diverse measures applied by heterogeneous individuals be mentioned as a limitation/source of heterogeneity in the discussion.

As by the time of publication, the results will be approaching 1 year old, the study would be enhanced by re-running the search for any new results since 31st December 2018.

Discussion Comment e) I'm afraid I still disagree that "Access to non-specialists would essentially reduce the treatment gap for children with autism, and ensure mental health for all across the globe [45].”

Discussion Comment h) adding the authors' response to this query to the discussion would enhance it.

Reviewer #3: Dear Author, I do not have any further comment on your manuscript. I appreciate it.

Best regards

The reviewer

Reviewer #4: The manuscript in my opinion is technically sound.

I am not very competent in statistics to comment on the accuracy. A statistical expert may comment.

All data is available for review.

The English is standard and intelligible. The language used is simple for the reader’s understanding. The flow of the text is also fine in my opinion.

The paper overall looks organised and good to go to me.

Reviewer #5: All my concerns have been fully elucidated, missing sections and analyses have been completed. Finally, comprehension errors have been corrected.

7. PLOS authors have the option to publish the peer review history of their article (what does this mean?). If published, this will include your full peer review and any attached files.

Reviewer #1: No

Reviewer #2: Yes: Roxanne Keynejad

Reviewer #3: Yes: Ladislav Hosak

Reviewer #4: No

Reviewer #5: No

---

## [Author Response · Author response to Decision Letter 1]

2 Oct 2019

Dear Professor Cheungpasitporn,

 My coauthors & I are very grateful to you for kind feedback on the manuscript. We believe the extensive feedback received had greatly improved the quality of the manuscript. We have now considered and responded to all the comments by worthy reviewer and revised our manuscript accordingly. 

We look forward to a favorable decision in due time. 

Best wishes,

Dr. Ahmed Waqas

Corresponding author

Reviewer #2

General feedback

Thank you for your responses to my previous review; the changes made have enhanced the manuscript. Please see some comments below which should be addressed before acceptance for publication.

Response

We are very grateful to you for the kind feedback which has greatly improved the quality of our work. We have now responded to all of your comments below and also revised our manuscript accordingly. 

Comment 1

I agree with Reviewer #1 that random-effects meta-analyses across the publication as a whole would be preferable and clearer to the reader. Would the authors consider following this suggestion?

Response

We have indeed used random effects analyses throughout the manuscript. The relative weights provided in the forest plots are random weights. In this context, we have removed the statement on use of random effects for analyses presenting I2 > 40%. 

Following statement has now been added: “Depending on the extent of heterogeneity, data were pooled together using either the fixed or random effects. Heterogeneity was considered significant at a cut off value > 40%. However, we applied random effects analysis for all of the outcomes because of heterogeneity in assessment of outcomes across included studies [17].”

Comment 2

Methods Comment c) When I suggested including the search term 'randomized' I intended this to refer to an alternative to RCT rather than an additional requirement, so this would not have decreased the number of results. The authors have not addressed my question about the very narrow search. I would suggest that the narrowness of the search and the number of databases be included in the discussion as limitations.

Response

The limitations related to narrowness of search and number of databases has been added in the limitations section. We have also added a sentence on using a more comprehensive search strategy encompassing different terms for RCT.’

Following sentences have been added: “The present systematic review was based on searching of a limited number of databases, we encourage investigators to search more databases in future studies. Moreover, investigators should also consider using a more comprehensive search strategy encompassing different terms for RCT.”

Comments

Methods Comment d) The authors have not clarified my query about whether the results represent a study of any interventions for participants of any age or whether in fact only studies of children were included. This needs to be clear throughout the manuscript.

Response

We have now added this as an exclusion criteria: “7. Interventions conducted among adults with ASD were excluded.”

Comment

Methods Comment g) I would suggest that combining results from diverse measures applied by heterogeneous individuals be mentioned as a limitation/source of heterogeneity in the discussion.

Response

We have now added following statement: “Combining results from diverse measures applied for heterogeneous study samples is another limitation of this systematic review.”

Comment

As by the time of publication, the results will be approaching 1 year old, the study would be enhanced by re-running the search for any new results since 31st December 2018.

Response

This is a very valid suggestion. However, due to limitation of resources such as funding and human resource, it would not be possible for us to re-run the searches and conduct more analyses. Hopefully, after publication of this manuscript, we will keep working on improving it. 

Comment

Discussion Comment e) I'm afraid I still disagree that "Access to non-specialists would essentially reduce the treatment gap for children with autism, and ensure mental health for all across the globe [45].”

Response

We have now deleted this statement from the manuscript.

Comment

Discussion Comment h) adding the authors' response to this query to the discussion would enhance it.

Response

We have now added following statements: Based on the findings of this systematic review, we cannot recommend one non-specialist mediated therapy for autism. PACT, PASS and PASS plus; JASPER, SENSE and Hanen’s more than words were tested in at least two studies and settings. Therefore, we recommend that future investigators, implementors and policy makers consult these therapy programs for development of interventions suitable for their settings.

---

## [Decision Letter · Decision Letter 2]

14 Oct 2019

Implementation and effectiveness of non-specialist mediated interventions for children with Autism Spectrum Disorder: A systematic review and meta-analysis

PONE-D-19-19450R2

Dear Dr. Ahmed Waqas,

We are pleased to inform you that your manuscript has been judged scientifically suitable for publication and will be formally accepted for publication once it complies with all outstanding technical requirements.

With kind regards,

Wisit Cheungpasitporn, MD, FACP, FASN

University of Mississippi Medical Center

Twitter: @wisit661 Email: wcheungpasitporn@gmail.com 

Academic Editor

PLOS ONE

Additional Editor Comments:

Your responses to the reviewer's comments were good and led you to make significant improvements to the paper.

Reviewers' comments:

Reviewer's Responses to Questions

**Comments to the Author**

1. If the authors have adequately addressed your comments raised in a previous round of review and you feel that this manuscript is now acceptable for publication, you may indicate that here to bypass the “Comments to the Author” section, enter your conflict of interest statement in the “Confidential to Editor” section, and submit your "Accept" recommendation.

Reviewer #1: All comments have been addressed

Reviewer #2: All comments have been addressed

Reviewer #3: All comments have been addressed

Reviewer #5: All comments have been addressed

2. Is the manuscript technically sound, and do the data support the conclusions?

Reviewer #1: Yes

Reviewer #2: Yes

Reviewer #3: Yes

Reviewer #5: Yes

3. Has the statistical analysis been performed appropriately and rigorously? 

Reviewer #1: Yes

Reviewer #2: Yes

Reviewer #3: Yes

Reviewer #5: Yes

4. Have the authors made all data underlying the findings in their manuscript fully available?

Reviewer #1: Yes

Reviewer #2: No

Reviewer #3: Yes

Reviewer #5: Yes

5. Is the manuscript presented in an intelligible fashion and written in standard English?

Reviewer #1: Yes

Reviewer #2: Yes

Reviewer #3: Yes

Reviewer #5: Yes

6. Review Comments to the Author

Reviewer #1: Everything looks good to me. I believe you should accept this manuscript.

Reviewer #2: Thank you for your attention to my additional comments. I am satisfied with all responses and am happy to recommend the current manuscript for publication.

Reviewer #3: Dear authors, I do not have any other comments on your manuscript. It seems to be O.K.

Best regards

The reviewer

Reviewer #5: It appears that all comments have been appropriately responded to. I have no further comments and recommend publication.

7. PLOS authors have the option to publish the peer review history of their article (what does this mean?). If published, this will include your full peer review and any attached files.

Reviewer #1: No

Reviewer #2: Yes: Dr Roxanne C Keynejad

Reviewer #3: No

Reviewer #5: No

---

## [Editor Report · Acceptance letter]

1 Nov 2019

PONE-D-19-19450R2 

Implementation and effectiveness of non-specialist mediated interventions for children with Autism Spectrum Disorder: A systematic review and meta-analysis 

Dear Dr. Waqas:

I am pleased to inform you that your manuscript has been deemed suitable for publication in PLOS ONE. Congratulations! Your manuscript is now with our production department. 

With kind regards,

on behalf of

Dr. Wisit Cheungpasitporn 

Academic Editor

PLOS ONE